# Pre-trained molecular representations enable antimicrobial discovery

Roberto Olayo-Alarcon [1,2] ✉, Martin K. Amstalden[3], Annamaria Zannoni [4], Medina Bajramovic [1,2], Cynthia M. Sharma [4], Ana Rita Brochado [3,5,6], Mina Rezaei[1] & Christian L. Müller[1,2,7] ✉

The rise in antimicrobial resistance poses a worldwide threat, reducing the efficacy of common antibiotics. Determining the antimicrobial activity of new chemical compounds through experimental methods remains time-consuming and costly. While compound-centric deep learning models promise to accelerate this search and prioritization process, current strategies require large amounts of custom training data. Here, we introduce a light-weight computational strategy for antimicrobial discovery that builds on MolE (Molecular representation through redundancy reduced Embedding), a self-supervised deep learning framework that leverages unlabeled chemical structures to learn task-independent molecular representations. By combining MolE representation learning with available, experimentally validated compound-bacteria activity data, we design a general predictive model that enables assessing compounds with respect to their antimicrobial potential. Our model correctly identifies recent growth-inhibitory compounds that are structurally distinct from current antibiotics. Using this approach, we discover de novo, and experimentally confirm, three human-targeted drugs as growth inhibitors of *Staphylococcus aureus*. This framework offers a viable, cost-effective strategy to accelerate antibiotic discovery.

The development of novel antibiotics is a priority given the widespread dissemination of pathogenic strains resistant to current treatments[1]. Novel therapeutic candidates are often first identified by screening large chemical libraries. However, the success of these screenings is limited, with a typical hit rate between 1 and 3%[2,3]. This issue is compounded by the high cost of the experiments owing to the large size of the chemical libraries being evaluated (from thousands to millions of molecules). Furthermore, the limited variability in these libraries makes it challenging to validate newly discovered or synthesized chemical species[2,4]. In this context, deep learning-assisted strategies

hold the promise to greatly contribute to the prioritization of molecules that should be experimentally evaluated, thus accelerating the rate at which novel drug candidates are found[5].

Using computational methods to estimate and predict properties of molecules, traditionally referred to as quantitative structure-activity relationships modeling, has a long history in material sciences, molecular biology, and biochemistry[6,7]. The success of these modeling approaches hinges on an appropriate representation of the molecules, i.e., chemical descriptors such as the Extended Connectivity Fingerprint (ECFP)[8], and the availability of sufficiently large training data to

[1]Department of Statistics, Ludwig-Maximilians-Universität München, Munich, Germany. [2]Institute of Computational Biology, Helmholtz Zentrum München, Munich, Germany. [3]Department of Microbiology, Julius-Maximilians-Universität Würzburg, Würzburg, Germany. [4]Department of Molecular Infection Biology II, Institute of Molecular Infection Biology (IMIB), Julius-Maximilians-Universität Würzburg, Würzburg, Germany. [5]Interfaculty Institute of Microbiology and Infection Medicine Tübingen (IMIT), University of Tübingen, Tübingen, Germany. [6]Cluster of Excellence 'Controlling Microbes to Fight Infections' (CMFI), University of Tübingen, Tübingen, Germany. [7]Center for Computational Mathematics, Flatiron Institute, New York, USA. ✉e-mail: roberto.olayo@lmu.de; christian.mueller@helmholtz-munich.de

map structure and activity. In recent years, the field of computational molecular property prediction has seen major breakthroughs, largely owing to advances in graph neural networks (GNNs)[9–12]. These graph-based methods adopt an end-to-end supervised learning framework where predictive models infer a task-specific latent representation from large-scale training data. Important application benchmarks include the Tox21 data[13], where twelve toxic effects of 12,000 environmental chemicals and drugs were measured and provided as prediction challenges, and the MoleculeNet collection[14] which comprises prediction tasks for 16 datasets spanning different application domains. In the context of antimicrobial discovery, a major focus has been the prediction of antimicrobial peptides (AMPs) using curated databases such as CAMP_{R3} (see[15] and references therein). Despite the success of sophisticated deep-learning architectures for AMP prediction[16–19], their task specificity limits the transferability of the learned representation to novel tasks and molecule types. Likewise, the application of Directed Message Passing Neural Networks (D-MPNNs)[12] for general antimicrobial discovery[2,20–22] required the creation of a custom training set via in-house compound screening. A large number of compounds (ranging from 2000 to 39,000 molecules) were tested for growth-inhibitory activity against each microbial

species of interest, requiring considerable lab expertise and resources. Despite recent advances in lab automation and analysis[23], a publicly available large-scale data resource for generic antimicrobial discovery is not yet available, thus hindering the straightforward use of end-to-end learning schemes.

In this contribution, we tackle the challenge of antimicrobial discovery by introducing a two-stage deep-learning strategy that enables the assessment of the antibacterial potential of any compound of interest (Fig. 1). The first stage uses a self-supervised pre-training strategy, termed MolE (Molecular representation through redundancy reduced Embeddings), for molecule representation (Fig. 1a). In the second stage we learn a set of antimicrobial potential (AP) scores for molecule prioritization that leverages the MolE representation and publicly available measurements of growth-inhibiting effects of FDA-approved drugs, including human-targeted and anti-infective drugs, against a diverse set of bacterial species[24] (Fig. 1b, c).

Inspired by recent molecular self-supervised learning schemes[25–28], MolE leverages the large collection of available unlabeled chemical structures from PubChem[29] to learn a general-purpose molecular representation that is transferable to downstream prediction tasks.

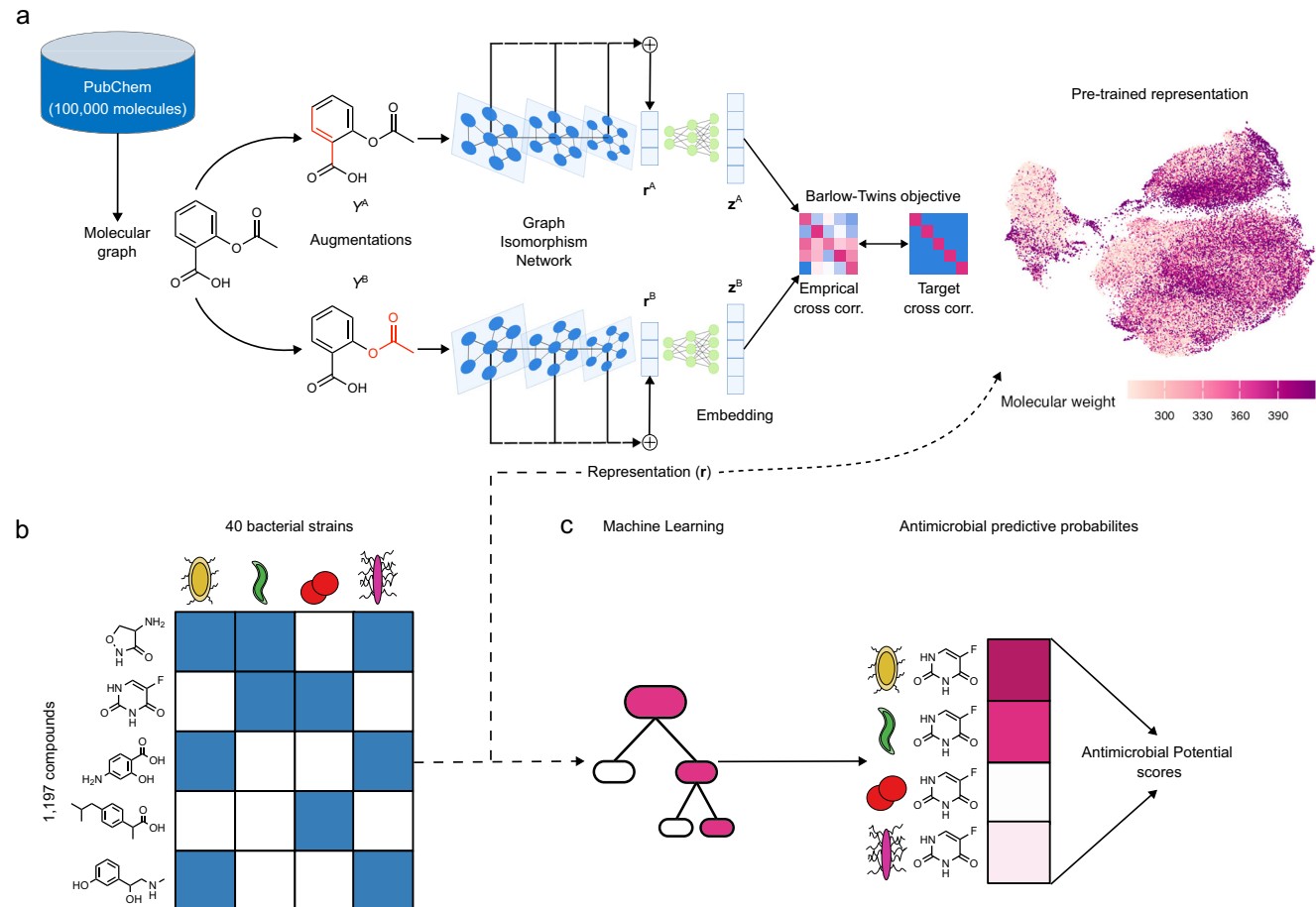

**Fig. 1 | Two-stage framework for antimicrobial discovery. a** The MolE pre-training framework uses a collection of 100,000 unlabeled structures from Pub-Chem to learn a task-independent, molecular representation. Each structure is represented as a molecular graph, from which two augmentations are created ($Y^A$ and $Y^B$) by masking a randomly seeded subgraph. Each augmentation is encoded by a GIN backbone to produce a concatenated vector representation ($r^A$, $r^B$) which is then expanded into embedding vectors ($z^A$, $z^B$) using an MLP head. The cross correlation between the two embedding vectors is optimized to be as similar as possible the target identity matrix using the Barlow-Twins objective function. After pre-training, any molecular structure can be encoded into a fixed-length vector representation **r**, which captures relevant chemical information. **b** Publicly available measurements of growth inhibition against 40 microbial strains[24] are used to train a predictive model. **c** The pre-trained molecular representation is combined with the compound-microbe activity measurements to train a machine-learning model that produces a probability for each compound-microbe combination, indicating how likely the compound is to inhibit the microbe's growth. These probabilities are used to estimate a collection of Antimicrobial Potential scores that serve to prioritize compounds for experimental validation.

MolE uses Graph Isomorphism Networks (GINs) for representation learning and introduces the non-contrastive Barlow-Twins pre-training framework[30] to the molecular domain. Combined with supervised learning schemes, MolE enables competitive predictive performance on a series of curated molecular property prediction tasks from MoleculeNet.

For the purpose of antimicrobial discovery, we show that MolE-derived AP scores not only reflect the broad- and narrow-spectrum activity of structurally diverse compounds, such as Halicin[2] and Abaucin[31], but can also serve as compound prioritization guide in large-scale chemical screens. On a separate chemical library of over 2000 compounds, we identified approximately 200 compounds with high AP scores and potential broad-spectrum activity. We observed significant associations between MolE-derived AP scores and experimentally measured minimum-inhibitory concentration (MICs) for a large set of the discovered non-antibiotic compounds. Among the set of predicted high-AP molecules with no known antimicrobial activity, we used our framework to prioritize six compounds for experimental growth-assay validation on four bacterial species. We confirmed significant inhibitory effects of three of the six compounds on the growth of the human pathogen *Staphylococcus aureus*, reaching an notable success rate compared to the state of the art[3].

We envision that the presented workflow and methodologies such as ours will be adopted by a wide range of microbiologists as a general and cost-effective way to prioritize and discover novel molecules with antibiotic properties.

## Results

### MolE learns meaningful compound representations

An important prerequisite for discovering compounds with potentially antimicrobial properties is the use of a general, yet efficiently tunable, representation of molecular structures. To provide such a context-independent representation we developed the MolE (Molecular representation through redundancy reduced Embedding) framework (Fig. 1a). MolE is a non-contrastive self-supervised deep learning scheme that constructs a representation of molecular structures by applying the Barlow-Twins redundancy reduction scheme[30]. We chose the Barlow-Twins framework because of its superior transfer learning capabilities and insensitivity to hyper-parameter choices when compared to its contrastive counterparts[30]. The input into our pre-training framework is based on SMILES (simplified molecular-input line-input system[32]), a popular text-based representation of chemical structures. The workflow, illustrated in Fig. 1a, comprises five main steps: (i) the SMILES representation of molecules is used to construct a molecular graph, where each node represents an individual atom and each edge a chemical bond in the molecule; (ii) two augmentations are created for each molecule by masking a subgraph of the original structure; (iii) batches of these augmentations enter a series of Graph Isomorphism Network (GIN) layers for feature extraction; (iv) the pooled output of each GIN layer is collected to form a final vector representation $\mathbf{r} \in \mathbb{R}^{1000}$; (v) a non-linear projection head expands each of the two vector representations into an embedding of higher dimensionality $\mathbf{z} \in \mathbb{R}^{D}$, serving as input to the Barlow-Twins objective function $\mathcal{L}_{BT}$ (Methods). After pre-training, the static MolE representation $\mathbf{r}$ can readily serve as input for downstream machine learning applications (such as Fig. 1b). Alternatively, concatenating the MolE architecture with an additional predictive layer allows fine-tuning of the pre-trained GIN layer parameters, thus making the representation adaptive to a specific downstream task.

To investigate MolE's pre-trained representation, we first examined similarities between the representations of 100,000 test molecules not seen during pre-training. Figure 2a shows a UMAP[33] embedding of these molecules using the MolE representation. We observe that MolE learned similar representations for molecules with matching functional groups and/or related topological features. For

instance, compounds consisting of a naphthalene group connected to a long carbon chain are placed closely in the embedding space (Fig. 2a top middle). Other examples include Pyridines bound to central nitrogen heterocycles (Fig. 2a top right), Benzenes surrounded by ether bonds (Fig. 2a lower right), and Nitrogen heterocycles with various decorations (Fig. 2a lower left). Notably, MolE's representation also recognizes the similarities in the structure of short amino-acid chains (Fig. 2a top left). Indeed, these peptides belong to a distinct cluster of molecules in the embedding space, indicating that MolE can distinguish this molecule type without any further fine-tuning. This offers an advantage over AMP-specific models by capturing broader chemical variability.

Compared to the standard ECFP4 representation (see Supplementary Fig. 1 for a corresponding UMAP embedding), MolE's representation captures distinct features for the same molecules. For illustration, we chose Ractopamine (PubChem ID: 56052, Fig. 2c) as a hypothetical query molecule. We extracted its corresponding MolE and ECFP4 representations and calculated similarities to all other test set molecules (shown in Fig. 2a) with the cosine and Jaccard (also known as Tanimoto) distances, respectively (Methods).

Figure 2b shows the high-level agreement between the two representations in terms of pair-wise distances to the query molecule (Spearman correlation: 0.42). However, discrepancies arise in the context of relevant nearest neighbors of the query, i.e., the most similar molecules to Ractopamine. Figure 2c ranks the top four most similar molecules in either representation. MolE's most highly ranked molecules share two phenol groups connected by a carbon chain with one amine functional group with the query, ECFP4 only one phenol ring and a carbon chain with a methyl functional group. This illustrates MolE's chemically meaningful embedding capabilities. Further examples are listed in Supplementary Fig. 2, where MolE was able to highlight global structural features such as a naphthalene group connected to a long carbon chain (Supplementary Fig. 2b), as well as the presence of specific functional groups such as sulfonyl chloride (Supplementary Fig. 2c).

Taken together, these observations show that MolE captures chemically relevant information from unlabeled molecular structures. It recognizes the presence of functional groups and learns similar representations for molecules that share structural characteristics. The similarities recovered by MolE are distinct from those uncovered by ECFP4, potentially boosting the performance in downstream prediction tasks.

### MolE enables competitive molecular property prediction

To showcase MolE's ability to achieve competitive molecular property prediction, we evaluated the out-of-sample performance of XGBoost[34] and Random Forests[35] when trained with MolE's static representation as well as feed-forward neural networks that fine-tune MolE's GIN layer weights. We considered selected classification and regression tasks from MoleculeNet[14] that are relevant for our context of compound prioritization in antimicrobial discovery, such as, e.g., the Tox21 and ClinTox benchmarks. Each dataset was split into training, validation, and testing sets following the scaffold splitting procedure[36], thereby creating a realistic scenario where molecules seen during training are structurally distinct from molecules seen during validation and testing. Performance is measured in terms of Area Under the Receiver Operating Characteristic curve (ROC-AUC) values obtained on the test set for each task.

We observed that MolE's static representation (MolE$_{static}$) combined with XGBoost outperformed alternative approaches in the majority of the classification and regression tasks (Table 1 and Supplementary Table 1, respectively). MolE$_{static}$ enabled the best metric in four out of six classification tasks, with an average performance increase of 3% (in terms of average ROC-AUC) compared to the ECFP4 representation. Notably, XGBoost with MolE outperformed

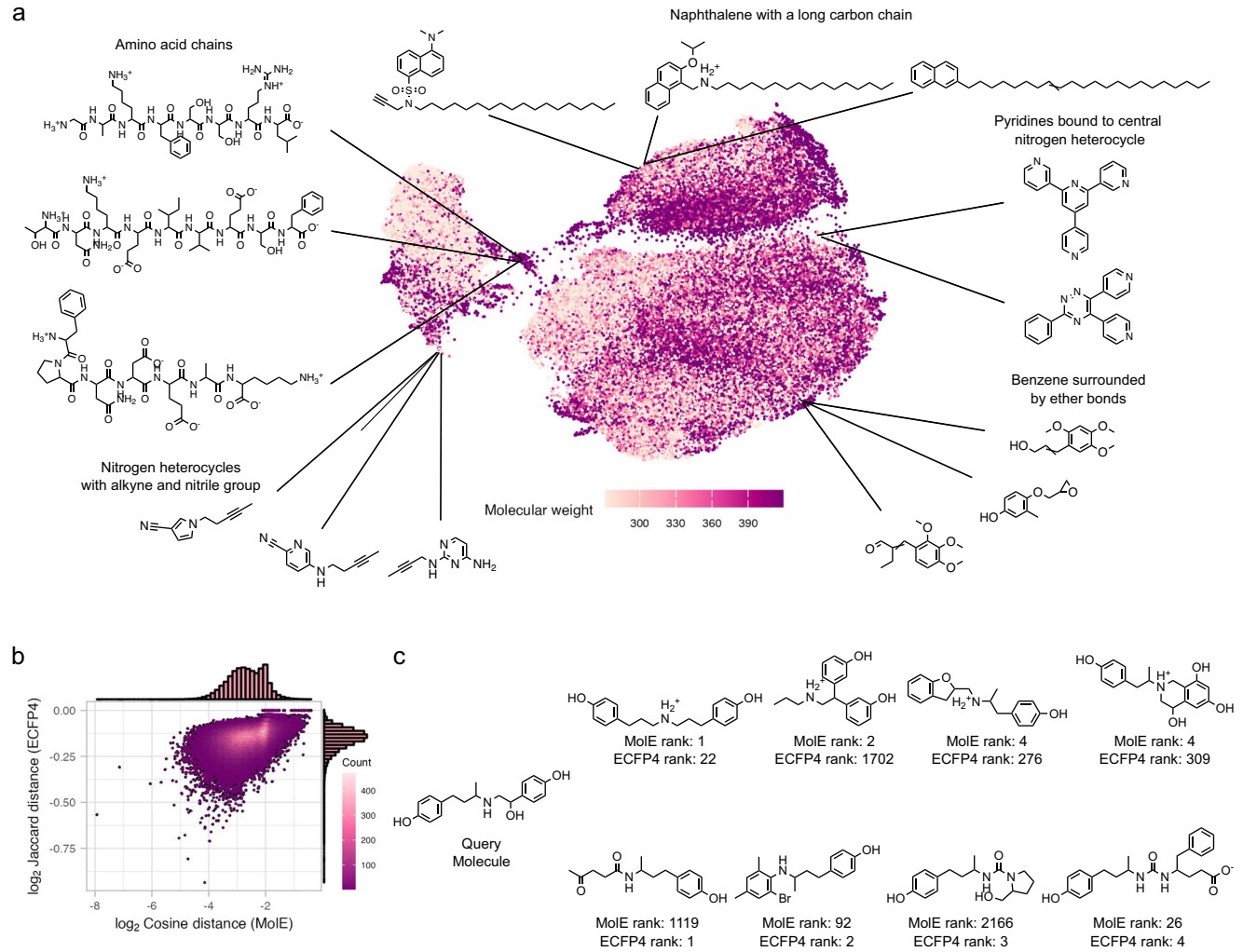

**Fig. 2 | Illustration of MolE's compound representation. a** UMAP embedding of MolE's representation of 100,000 chemical structures not seen in pre-training. **b** Comparison of Jaccard distance computed between ECFP4 representations and cosine distance computed between MolE representations with respect to the query molecule Ractopamine (PubChem ID: 56052), shown in panel **c**. Comparison of the four molecules with smallest distances (ranks) to Ractopamine according to MolE (top row) and ECFP4 (bottom row).

the end-to-end learning scheme D-MPNN in four classification tasks and two regression tasks. The added benefit of using the MolE$_{static}$ representation was not limited to XGBoost as we observed similar performance gains with Random Forest predictors (Supplementary Tables 2 and 3). In general, fine-tuned MolE-based predictions (MolE$_{finetune}$) displayed similar performance to their static counterparts. On the ClinTox classification benchmark, however, MolE$_{finetune}$ achieved an average ROC-AUC of 92.85%, considerably outperforming all other approaches and even rivaling fine-tuned classifiers on large-scale ChemBERTa (90.6%) or MoLFormer (94.8%) representation models[25,26]. This confirms that the computationally light-weight MolE framework can learn meaningful molecular representations with only ~100,000 unlabeled structures from PubChem (Supplementary Fig. 2d), contrasting the large-scale pre-training schemes ChemBERTa[25] or MoLFormer[26], requiring millions of examples. To identify the contributing factors of this performance improvement, we conducted an ablation study of all MolE components, including the use of the Barlow-Twins objective function, pretraining sample sizes, and representation dimensions (Methods and Supplementary Fig. 3). Finally, the benchmarks and the ablation study also revealed MolE's pre-training architecture, in combination with the non-contrastive Barlow-Twins strategy, to be superior to the state-of-the-art contrastive MolCLR framework (Table 1 and Supplementary Fig. 3e).

Taken together, the benchmarking results indicate that (i) MolE produces a molecular representation that enables excellent downstream prediction performance even when pre-trained on a small dataset of 100,000 unlabeled structures (Supplementary Fig. 3d) and (ii) MolE is particularly competitive for downstream tasks with a small number of labeled data (such as ClinTox and BACE).

## MolE enables generalizable predictions of antimicrobial compound activity against human gut microbes

The second stage of our framework (see Fig. 1b, c) leveraged MolE's representation capabilities to learn a set of antimicrobial potential scores that allow ranking and prioritization of compounds in the antimicrobial discovery process. To this end, we made use of the publicly available dataset created by Maier et al.[24], which evaluated the effect of 1197 marketed drugs on the growth of 40 bacterial strains representative of the human gut microbiome (Fig. 1b). This dataset was used to train an XGBoost model to learn the probability of growth inhibition for all available compound-microbe pairs (Fig. 3a, b). For any compound of interest, the trained model, termed MolE-XGBoost, delivers a 40-dimensional vector of predictive probabilities (Fig. 3b). Data-driven thresholding of these probabilities allows for (i) binary classification of a compound to be growth-inhibitory of a specific species or (ii) assessment of narrow- or broad-spectrum activity of the

**Table 1 | Average ROC-AUC (%) and standard deviation obtained on classification benchmark tasks**

| Performance on Classification Tasks | | | | | | |
|---|---|---|---|---|---|---|
| (Higher is better) | | | | | | |
| Dataset | BBBP | Tox21 | ClinTox | BACE | SIDER | HIV |
| # Molecules | 2039 | 7831 | 1478 | 1513 | 1427 | 41127 |
| # Tasks | 1 | 12 | 2 | 1 | 27 | 1 |
| GCN[27] | 71.8 ± 0.9 | 70.9 ± 2.6 | 62.5 ± 2.8 | 71.6 ± 2.0 | 53.6 ± 3.2 | 74.0 ± 3.0 |
| GIN[27] | 65.8 ± 4.5 | 74.0 ± 0.8 | 58.0 ± 4.4 | 70.1 ± 5.4 | 57.3 ± 1.6 | **75.3 ± 1.9** |
| SchNet[27] | 84.8 ± 2.2 | **77.2 ± 2.3** | 71.5 ± 3.7 | 76.6 ± 1.1 | 53.9 ± 3.7 | 70.2 ± 3.4 |
| MGCN[27] | **85.0 ± 6.4** | 70.7 ± 1.6 | 63.4 ± 4.2 | 73.4 ± 3.0 | 55.2 ± 1.8 | 73.8 ± 1.6 |
| D-MPNN[27] | 71.2 ± 3.8 | 68.9 ± 1.3 | **90.5 ± 5.3** | **85.3 ± 5.3** | **63.2 ± 2.3** | 75.0 ± 2.1 |
| ECFP4 | 68.51 ± 1.01 | 70.23 ± 0.76 | 84.52 ± 0.00 | **84.33 ± 1.12** | 62.58 ± 1.89 | 75.77 ± 1.6 |
| N-Gram | 74.00 ± 0.00 | 73.51 ± 0.54 | **89.68 ± 1.27** | 81.97 ± 0.00 | 63.68 ± 1.59 | **78.89 ± 0.48** |
| MolCLR$_{static}$ | 65.74 ± 0.68 | 72.05 ± 1.34 | 76.04 ± 3.05 | 68.78 ± 0.00 | 62.68 ± 2.40 | 74.29 ± 0.00 |
| MolE$_{static}$ | **75.98 ± 0.32** | **76.15 ± 0.61** | 84.90 ± 0.00 | 83.84 ± 0.86 | **64.48 ± 2.65** | **78.52 ± 0.00** |
| Hu et al.[27] | 70.8 ± 1.5 | **78.7 ± 0.4** | 78.9 ± 2.4 | **85.9 ± 0.7** | 62.7 ± 0.8 | **80.20 ± 0.9** |
| HiMol[54] | 73.2 ± 0.80 | 76.2 ± 0.30 | 80.8 ± 1.4 | 84.6 ± 0.20 | 62.5 ± 0.30 | 74.71 ± 0.23 |
| MolCLR$_{finetune}$ | 73.17 ± 2.13 | 74.61 ± 1.67 | 87.79 ± 5.31 | 81.96 ± 1.11 | 63.23 ± 3.05 | 77.61 ± 0.92 |
| MolE$_{finetune}$ | **75.34 ± 0.93** | 75.21 ± 1.77 | **92.85 ± 2.46** | 83.21 ± 1.84 | **64.98 ± 4.42** | 78.98 ± 0.57 |

The first 5 models are supervised learning methods. The next 4 are the names of molecular features given as input to an XGBoost classifier. The final four methods are fine-tuned models. The best performance metric for each category is marked in bold.

compound by inspecting the overall number of inhibited strains, respectively (Fig. 3c, Eq. (11)).

MolE-XGBoost strongly outperformed ECFP4-based models and a recently proposed predictive model[4] that uses a collection of explicit chemical descriptors in terms of Precision and Recall (Fig. 3d). Furthermore, MolE-XGBoost accurately predicted the broad-spectrum activity of the human-targeted drug Diacerein, which was a part of the test set in our experiment (Fig. 3e). The model correctly predicted an effect on 25 of the 33 strains that showed inhibited growth in lab experiments (Fig. 3e bottom row). Notably, the ECFP4-XGBoost model failed to recover any antimicrobial activity for the same drug. Similar examples are highlighted in Supplementary Fig. 4.

We next confirmed the generality of MolE-XGBoost by recapitulating the findings from recent ground-breaking studies that identified structurally novel antibiotic candidates[2,31]. Firstly, our MolE-XGBoost model was able to re-discover the broad-spectrum activity of Halicin[2] (Fig. 3f, bottom row) and correctly predicted the experimentally validated inhibition of *Escherichia coli* and *Clostridium difficile*.

Secondly, in the case of Abaucin[31], the model recovered its highly narrow-spectrum activity, predicting only the (yet to be tested) inhibition of *Eubacterium rectale* (Fig. 3f bottom row).

To further highlight the superior performance of MolE-based predictions, we focused on the 24 compounds in the test set that had experimentally determined broad-spectrum antimicrobial activity (i.e., inhibiting the growth of ten or more strains[24]). Figure 3g reports the true number of species affected by each compound (last row) vs. Chemical Descriptor-based, ECFP4-based, and MolE-based predictions, respectively (top three rows). On average, MolE-XGBoost achieved the highest accuracy in recovering the measured antimicrobial activity of compounds independent of the intended target of the compound. The ECFP4-based model failed to recall the activity of most human-targeting drugs, while the Chemical Descriptor-based model generally overestimated activities, leading to many false positive predictions (Fig. 3g and Supplementary Fig. 5a). MolE-XGBoost recovered five compounds with broad-spectrum activity, not recognized by the ECFP4-based model.

## Predicting antimicrobial potential scores in an orthogonal chemical library

Next, we used MolE-XGBoost's predictions to identify de novo bacterial growth inhibitors in an orthogonal chemical library of 2,320 FDA-approved drugs, food homology products, and human endogenous metabolites from MedChemExpress (MCE). This new discovery MCE library contained compounds not seen during model training and covered a broader chemical space (see Supplementary Fig. 6 for a joint UMAP embedding of the discovery library and the previous training data).

We used the model's predictive probabilities for each of the 40 bacterial strains to design four variants of Antimicrobial Potential (AP) scores that allow the ranking of individual compounds:

(i) The total number of strains predicted to be inhibited $K$ (Fig. 3c, Eq. (10)),

(ii) The $\log_2$-geometric mean of estimated probabilities across all 40 strains $G$ (Fig. 3b, Eq. (12)),

(iii) The $\log_2$-geometric mean of the probabilities of all 22 gram-positive strains $G^+$ (Eq. (13)),

(iv) The $\log_2$-geometric mean of the probabilities of all 18 gram-negative strains $G^-$ (Eq. (14)).

Following the operational definition in[24], we consider a compound to be a potential broad-spectrum inhibitor of microbial growth if it is predicted to inhibit the growth of ten or more strains ($K \geq 10$). Here, we prioritize compounds according to this broad-spectrum definition and use the Gram-based AP scores $G^+$ and $G^-$ to assess compound target specificity. We remark that our antimicrobial potential scores are expected to encompass a broad range of effects, ranging from growth-delaying or -limiting effects all the way to complete inhibition.

Figure 4a illustrates the diversity of the library's 2320 compounds in a MolE-based UMAP embedding, highlighting the 235 predicted broad-spectrum inhibitor compounds. Among these, 158 compounds were non-antibiotics. Our extensive post-hoc literature review revealed that, out of these 158 compounds, 53 had previously been reported to inhibit the growth of various bacterial species (Fig. 4b). Examples include structurally diverse compounds such as natural products Ellagic acid[37] and Shionone[38], as well as human-targeted

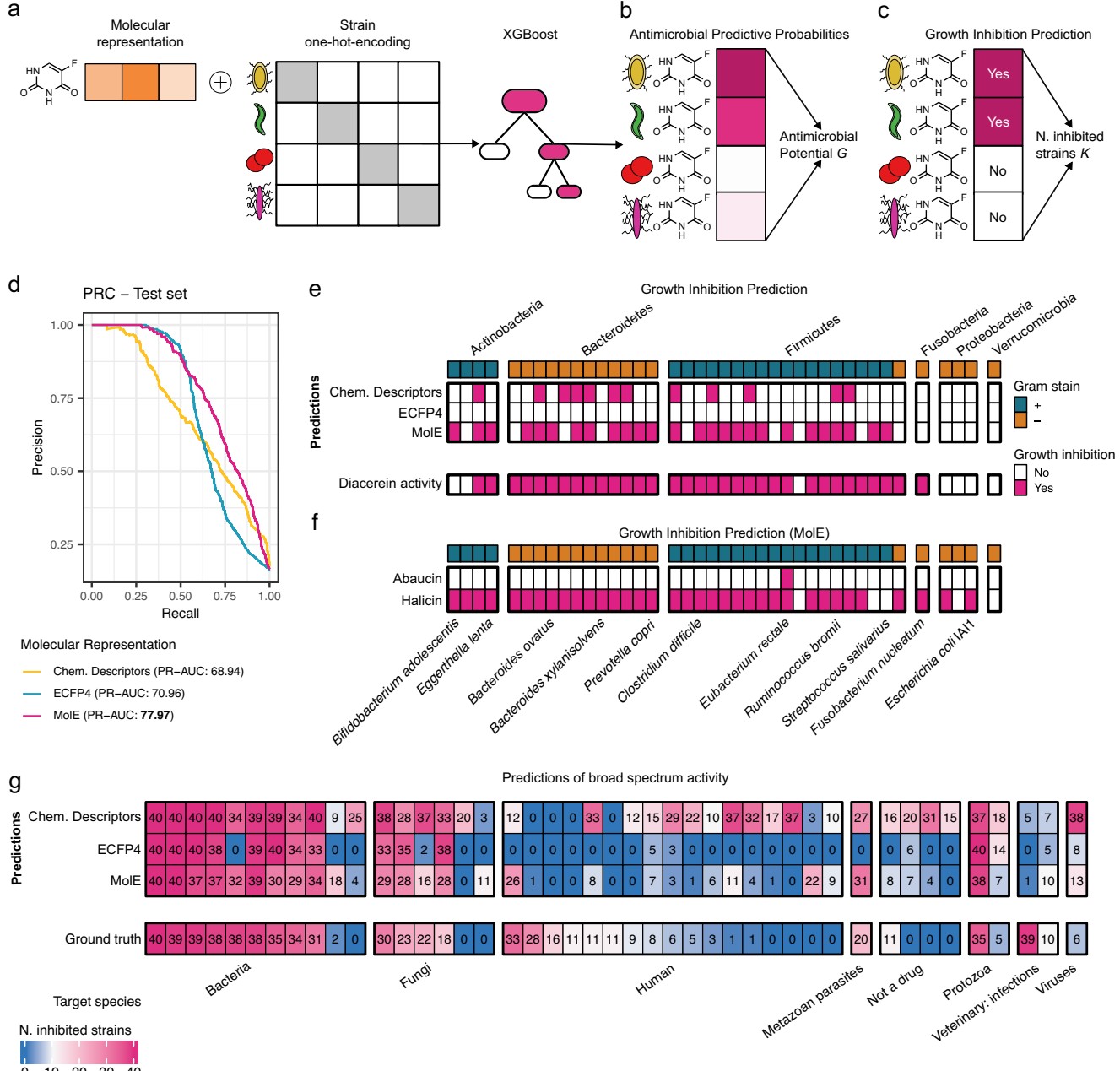

**Fig. 3 | Predicting antimicrobial activity in the human gut microbiome.**
**a** Molecular representations (ECFP4, Chemical Descriptors or MolE) are con-catenated with a one-hot-encoding of the microbial strains to train an XGBoost model. **b** The model produces an antimicrobial predictive probability for each compound-microbe pair. The log-geometric mean of all probabilities corresponds to AP score *G*. **c** The predictive probabilities are thresholded into binary predictions of growth inhibition; the total number of strains predicted to be inhibited is determined (*K*). **d** Precision-recall curves on the test set for models trained with each molecular representation. PR-AUC is shown in legend. **e** Binary predictions for

the growth-inhibitory activity of Diacerein by each model. The experimentally validated activity is shown in the bottom row. Each column is an individual strain. **f** Predictions for the antimicrobial activity of Halicin and Abaucin made by MolE-XGBoost. **g** List of 44 compounds in the test set comprising 24 compounds with experimentally validated broad-spectrum activity (i.e., inhibited strains ≥10) grouped by their intended target species[24]. Each column represents a compound. Each entry represents the number of predicted inhibited strains (color-coded). The last row represents the experimentally determined ground truth number of inhibited strains.

drugs such as Doxifluridine[39,40] (Fig. 4a). This putative 33% hit rate considerably improves the 1–3% rate commonly cited for large-scale chemical screens[3].

We further assessed the validity of our proposed AP scores by testing their ranking capabilities on the 93 known antibiotics present in the chemical library. Figure 4c shows the distribution of general AP score *G* vs. the predicted number of inhibited strains *K* for all com-pounds. We observed strong ranking capability, with an enrichment of

antibiotic compounds at large AP scores (Fig. 4c). Furthermore, 77 of the 93 antibiotics were predicted to inhibit 10 or more strains (Fig. 4c, dashed line), confirming that both scoring schemes generalize well to unseen antibiotic compounds.

Finally, we assessed the discriminative potential of the refined AP scores for Gram-positive and Gram-negative strains by simultaneously ranking the 158 non-antibiotic compounds previously predicted to be broad-spectrum (Fig. 4d). In line with current knowledge about the

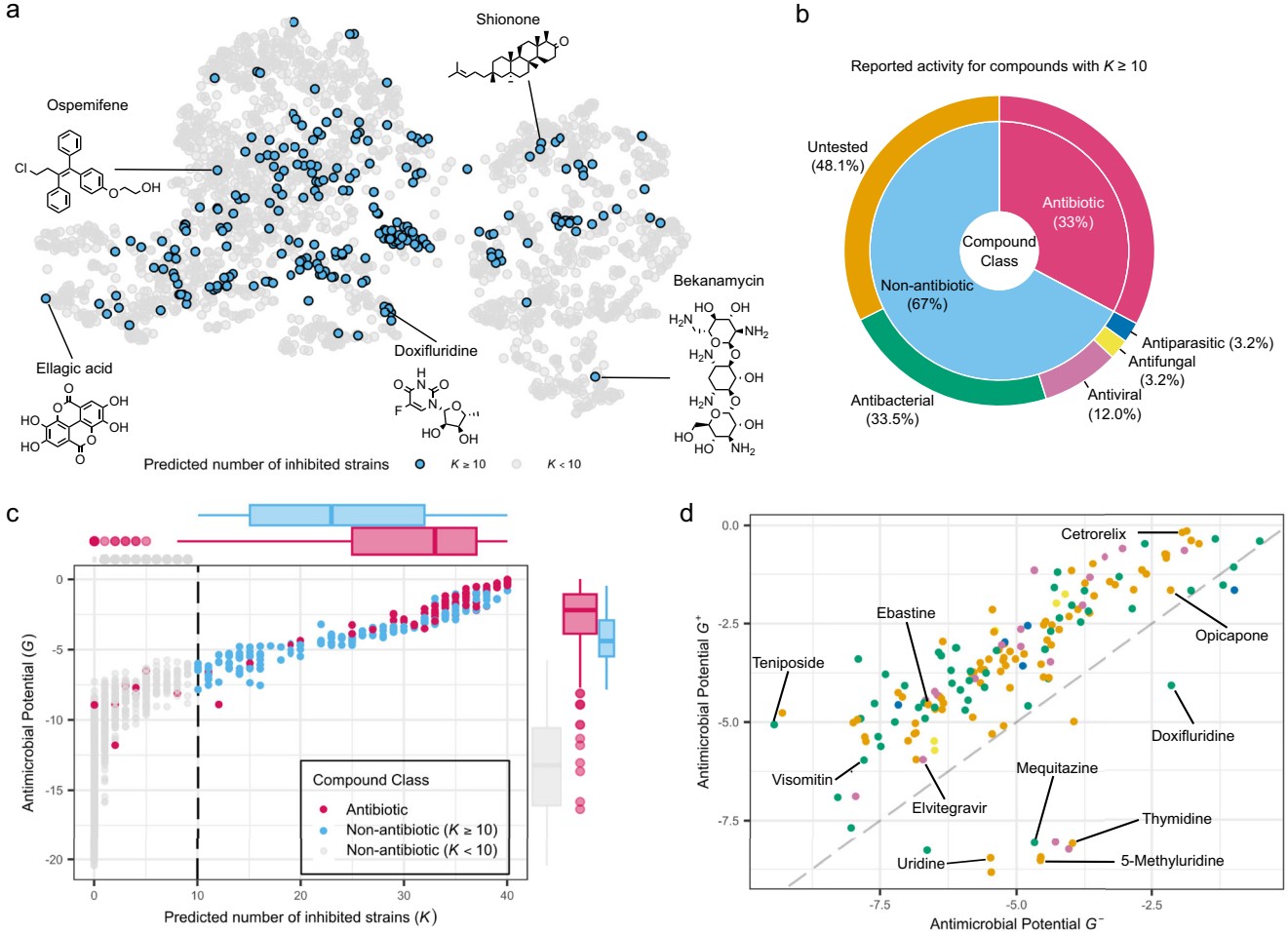

**Fig. 4 | Predicting antimicrobial potential in the discovery MCE-based chemical library comprising 2320 compounds. a** UMAP embedding of MolE's representation of the 2320 compounds for which predictions are made. Compounds predicted to inhibit at least 10 strains ($K \geq 10$) are highlighted in blue. **b** Literature-reported categorization of the antimicrobial activity for the 235 compounds with $K \geq 10$. This set comprises 77 antibiotics (33%, shown in red) and 158 non-antibiotic drugs (67%, shown in blue). The non-antibiotic drugs are further categorized into five classes (colored in the outer ring). **c** Antimicrobial Potential score $G$ vs. number of predicted inhibited strains $K$ of all 2320 compounds. The dashed line marks $K = 10$. All known antibiotics present in the library ($n = 93$ antibiotics) are shown in red, while non-antibiotic compounds with $K \geq 10$ ($n = 158$ compounds) are shown in blue. Boxplots show the distribution of $G$ and $K$, with the median value shown as the middle line, first (Q1) and third quartiles (Q3) shown as the box limits, and the whiskers extending to the most extreme data points within 1.5 times the

interquartile range. The top boxplots show the distribution of $G$ for antibiotics ($n = 93$, median = −2.18, Q1 = −3.86, Q3 = −1.05, lower whisker = −7.71, upper whisker = −7.4 × 10⁻⁵), non-antibiotics with $K \geq 10$ ($n = 158$, median = −4.38, Q1 = −5.47, Q3 = −2.91, lower whisker = −7.83, upper whisker = −0.45), and non-antibiotics with $K < 10$ ($n = 2068$, median = −13.27, Q1 = −16.09, Q3 = −10.56, lower whisker = −20.43, upper whisker = −5.73). Similarly, boxplots on the right show the distribution of $K$ for antibiotics (median = 33, Q1 = 25, Q3 = 37, lower whisker = 8, upper whisker = 40), non-antibiotics with $K \geq 10$ (median = 23, Q1 = 15, Q3 = 32, lower whisker = 10, upper whisker = 32), and non-antibiotics with $K < 10$ (median = 0, Q1 = 0, Q3 = 0, lower whisker = 0, upper whisker = 0). **d** Scatter plot of the Antimicrobial Potential for Gram-positive ($G^+$) vs. Gram-negative strains ($G^-$) determined for the 158 non-antibiotic drugs with predicted broad-spectrum activity ($K \geq 10$). The coloring of each compound corresponds to the categorization in (**b**).

susceptibility of Gram-positive bacteria to chemical stressors[24,41,42], we confirmed that most compounds exhibited greater AP scores against Gram-positive strains compared to Gram-negative strains (Fig. 4d, above the dashed line). However, we observed that nucleotide analogs, such as Uridine and Uridine derivatives, were predicted to be more active against Gram-negative strains (Fig. 4d, below the dashed line), confirming recent evidence that Uridine molecules are powerful adjuvants for the activity of aminoglycosides against *Escherichia coli*[43].

For completeness, we performed the same analysis using AP scores derived from the ECFP4-XGBoost model. Briefly, AP scores from the ECFP4-XGBoost model deemed (i) fewer compounds to have broad-spectrum activity (Supplementary Fig. 7), (ii) fewer known antibiotics to be inhibitory (Supplementary Fig. 8a), and (iii) only 82 non-antibiotic compounds to have broad-spectrum activity (vs. 158 from MolE-XGBoost AP scores). Of these, a lower proportion were

found to have growth-inhibiting activity against microbial life in the literature (Supplementary Fig. 8c). MolE-XGBoost AP scores recovered more non-antibiotic compounds with confirmed activity, 58 of which were not prioritized by ECFP4-XGBoost AP scores. ECFP4-XGBoost prioritized 13 compounds that are not present in the MolE-XGBoost-derived set (Supplementary Fig. 8d).

**Antimicrobial Potential scores correlate with experimentally determined minimum inhibitory concentrations**

We next investigated the relationship between MolE-XGBoost AP scores $G$, $G^+$, and $G^-$ and quantitative experimental data of bacterial growth inhibition. To this end, we performed an extensive literature search and collected all available data on in vitro measured minimum inhibitory concentrations (MICs) among the 158 non-antibiotic compounds predicted to have broad-spectrum activity. We identified 31

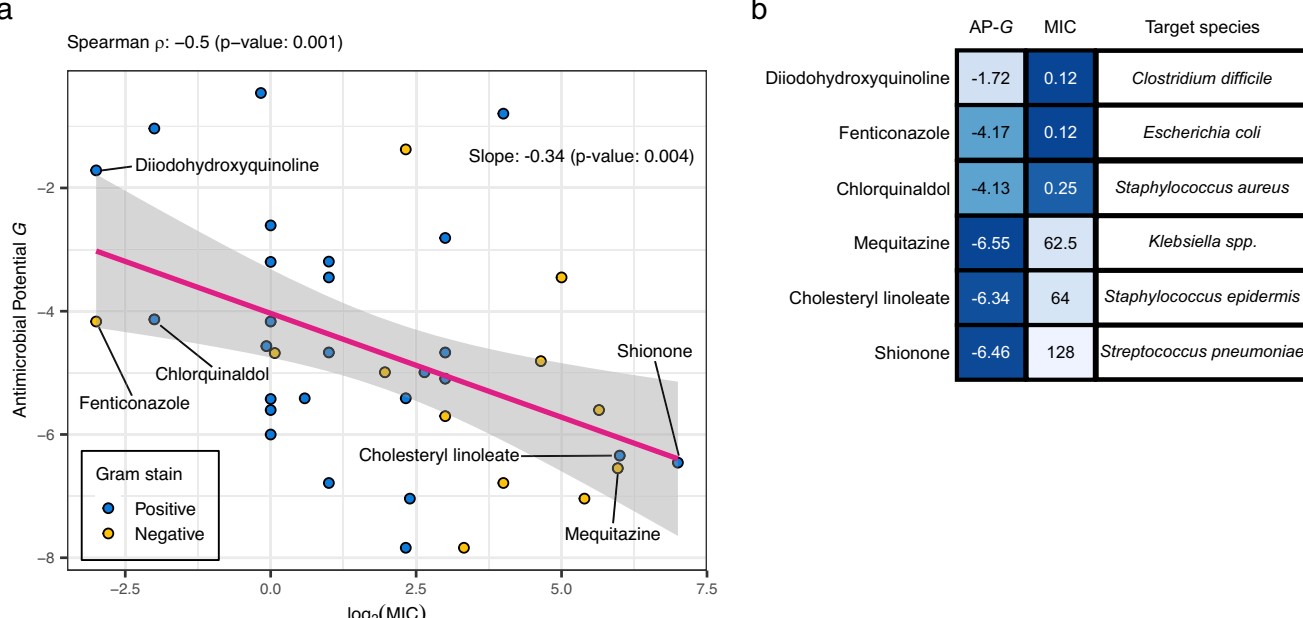

**Fig. 5 | Relationship between AP scores and minimum inhibitory concentration (MIC). a** Regression and correlation analysis between the AP score $G$ and the corresponding $\log_2$ literature-reported MICs ($\mu$g/mL) for 31 non-antibiotic compounds against Gram-positive (shown in blue) and Gram-negative (shown in yellow) species. In total, 39 compound-species combinations are shown, Spearman's $\rho = -0.5$ (two-sided correlation test $p$ value = 0.001). The linear regression fit is shown in pink with standard error shown as gray bands (slope = $-0.34$, two-sided coefficient test $p$ value = 0.004). **b** AP scores (AP-$G$) and MIC ($\mu$g/mL) values of the compounds with top-3 and bottom-3 MIC values along with the respective inhibited bacterial species.

compounds with an experimentally determined MIC ≤128 μg/mL against a wide variety of Gram-positive and Gram-negative species (see Supplementary Table 10). Figure 5a shows the relationship between MolE-XGBoost AP score $G$ and the experimentally measured MICs for both Gram-positive (marked in blue) and Gram-negative (marked in yellow) species. We observe a significant negative correlation (Spearman's $\rho = -0.5$), indicating that compounds with a high AP score have a more potent growth-inhibiting activity (i.e., a lower MIC). Figure 5b reports compounds with the top and bottom three MIC values and the corresponding inhibited species, together with MolE-XGBoost' AP scores $G$. Similar relationships hold when analyzing the Gram-specific AP scores $G^+$, and $G^-$ vs. MICs for the set of Gram-positive (Spearman's $\rho = -0.41$) and Gram-negative bacteria (Spearman's $\rho = -0.43$), respectively (see Supplementary Fig. 9a, b).

Notably, the experimentally determined MIC values were reported for several bacterial species that were not present in our training set, including *Streptococcus pneumoniae*, *Staphylococcus aureus*, and *Klebsiella*. This further confirms the ability of MolE-XGBoost AP scores to generalize to unseen compound-species pairs (Fig. 5 b).

Strikingly, when performing the same analysis for the model trained with ECFP4, we found no significant relationship between ECFP4-XGBoost AP scores and corresponding MICs (Supplementary Fig. 9c). Generally, we also found fewer compounds with a reported MIC ≤128 μg/mL when using the ECFP4-derived set of high-AP compounds (Supplementary Fig. 9d), including fewer non-antibiotic compounds active against Gram-negative and Gram-positive species (Supplementary Fig. 9e).

Taken together, the associations found between MolE-XGBoost predictions and external experimental data provide further evidence that our framework can identify and rank potent antibacterial compounds.

## Experimental validation of predicted antibacterial compounds

Guided by MolE-XGBoost AP scores on the discovery MCE library, we next selected six compounds to be experimentally validated via a MIC

assay: Cetrorelix, Ebastine, Elvitegravir, Opicapone, Thymidine, and Visomitin. The selected compounds (i) were predicted to be broad-spectrum ($K \geq 10$), (ii) had AP scores of $G^+ > -10$ and $G^- > -10$, (iii) covered a variety of functions and chemical structures (Fig. 4d), and (iv) were commercially available. Importantly, the chosen compounds were all structurally distinct from the antibiotics in our training set (Tanimoto similarity ≤0.3). Furthermore, Cetrorelix, Elvitegravir, Opicapone, and Visomitin shared low molecular similarity to all compounds seen during model training (Supplementary Fig. 10). With the exception of Visomitin, none of the chosen compounds had not been previously reported to inhibit the growth of bacterial or fungal strains. As a positive control, we included Visomitin as a non-antibiotic drug with proven antimicrobial activity[44].

We screened these compounds against a panel of bacterial strains that were selected to cover Gram-negative and Gram-positive pathogens, most part of the ESKAPE group[45]. These included the Gram-negatives *Escherichia coli* UTI, *Klebsiella pneumoniae*, *Pseudomonas aeruginosa*, and the Gram-positive *Staphylococcus aureus*, all of which were not present in the training dataset[24]. We also included one shared commensal strain, namely E. coli IAI1 as a representative of the human gut microbiome. The Gram-negative pathogens were taxonomically related to microbes from the Proteobacteria phylum in the training data, whereas *S. aureus* belonged to the Firmicutes phylum (Supplementary Fig. 11).

Overall, three of the six tested compounds were confirmed to have measurable effects on the growth of the Gram-positive pathogen *S. aureus*. The strongest effect was observed for Elvitegravir, which inhibited the growth of *S. aureus* at a concentration of 8 μg/mL (Fig. 6a). Furthermore, Opicapone significantly limited the growth of *S. aureus* to a maximum optical density of 0.38 ± 0.01 at a concentration of 128 μg/mL (Fig. 6b). At a lower concentration of 16 μg/mL, it extended both, the duration of the lag phase, and the population doubling time by about 1 h compared to growth observed with DMSO exposure (Supplementary Fig. 12b, c). Finally, Ebastine extended the duration of the lag phase of *S. aureus* by approximately 2 hours (Fig. 6c

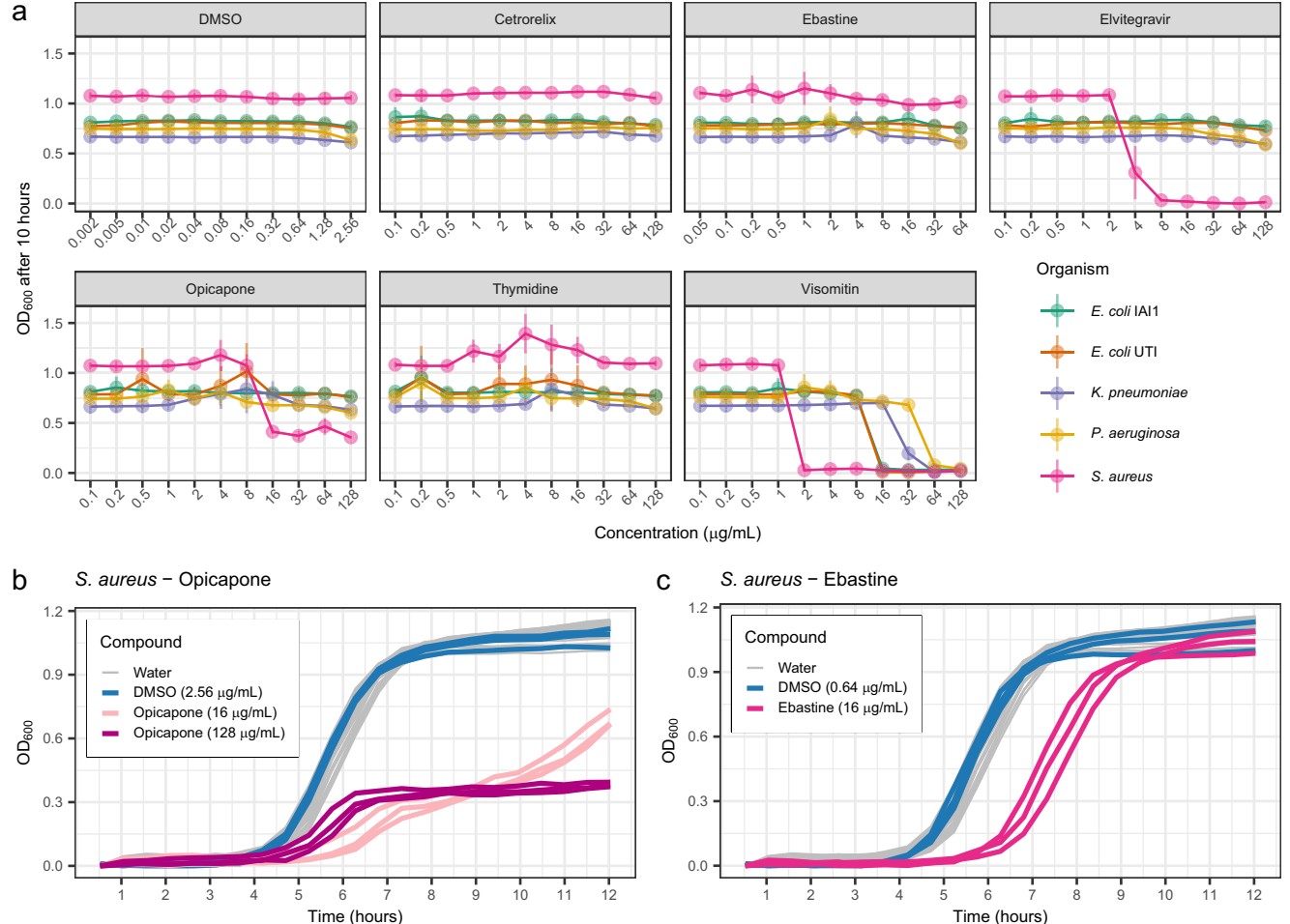

**Fig. 6 | Experimental validation of antimicrobial activity. a** Average OD$_{600}$ measurements ( ± standard deviation) after 10 hours of growth at increasing concentrations of each compound. Three biological replicates were performed for each compound-concentration-species combination. **b** Growth curves for *S. aureus* when grown in the presence of Water (33 growth curves gathered from 3 biological replicates), DMSO (3 biological replicates), and Opicapone (3 biological replicates for each concentration). **c** Growth curve for *S. aureus* when grown in the presence of Water (33 growth curves gathered from 3 biological replicates), DMSO (3 biological replicates), and Ebastine (3 biological replicates).

and Supplementary Fig. 13c). Notably, our experiments re-discovered the broad-spectrum activity of Visomitin, showing that it can inhibit the growth of *P. aeruginosa* and *K. pneumoniae* at 64 µg/mL, thereby expanding the list of species known to be susceptible to this compound[44].

While we did not observe growth inhibition in response to Cetrorelix and Thymidine, their predicted Antimicrobial Potential can be explained by the presence of similar molecules that affect the growth of bacteria in the training set such as, e.g., Azidothymidine. This medication is a chemical analog of Thymidine and was previously shown to inhibit the growth of twelve strains (nine Gram-negative and three Gram-positive)[24]. As a result, Thymidine was predicted to be particularly effective against Gram-negative strains by our model (Fig. 4d). For comparison, we also included the predicted antimicrobial effects of ECFP4- and Chemical Descriptor-based models for the five compounds in Supplementary Fig. 14.

In summary, the experimental validation demonstrated a notable discovery rate of our proposed MolE-based workflow, given the small number of compounds screened. The compounds found to affect the growth of *S. aureus* were structurally distinct from antibiotics present in the training data, a key feature desired for novel antimicrobials. While Opicapone and Ebastine did not completely inhibit *S.aureus*, they did exhibit measurable limits and delays on the standard growth dynamics. The molecular structure of these compounds can serve as a

starting point to explore chemical modifications that increase their potency[1]. Finally, the varied effects on bacterial growth observed for each compound suggest different mechanisms of action that can be further investigated in future studies.

## Discussion

In this contribution, we have presented (i) a computationally light-weight, predictive end-to-end workflow to identify novel anti-microbial candidates and (ii) experimentally confirmed growth-inhibitory effects of several compounds at a notable hit rate. Our framework specifically addresses the pervasive scarcity of biological and chemical data in the antimicrobial discovery process by leveraging the vast amounts of unlabeled chemical structures to learn a novel molecular representation within the MolE framework (Fig. 1a). This pre-trained representation captures relevant chemical and structural features (Fig. 2) and improves the performance of down-stream machine learning algorithms when predicting molecular properties when few labeled examples are available (Table 1 and Supplementary Table 1). The ability to obtain competitive perfor-mance from MolE's representation enables microbiology researchers with variable access to high-performance computing resources to make meaningful predictions of their property of interest, thus helping to democratize research molecular property prediction in microbiology, and beyond.

We have shown that by using MolE's pre-trained representation, standard XGBoost prediction models enable concise assessment of the antimicrobial potential of chemical compounds on publicly available data (Fig. 3). Our MolE-XGBoost model re-discovered de novo structurally distinct antibiotic candidates such as Halicin and recognized the broad-spectrum activity of other compounds that would have been missed by standard models. By designing and validating a set of model-derived antimicrobial potential (AP) scores, we have been able to provide a reliable ranking of the growth-inhibiting potential of any compound of interest, not only for specific microbial strains but also for the class of Gram-positive or Gram-negative bacteria, and for bacteria in general (Figs. 4, 5, 6). This allows a much broader applicability of our framework in the antibiotics discovery process.

While large-scale (blind) experimental screening of chemical libraries will remain a common technique for drug discovery, it is unlikely to keep pace with the continuous introduction of novel chemical species. Therefore, methodologies such as ours are a practical alternative to prioritize molecules for screening. We have demonstrated these potential efficiency gains with the experimental validation of our model predictions, where three out of six compounds were found to have significant effects on the growth of *S. aureus* (Fig. 6). These effects ranged from total growth inhibition (Elvitegravir) to delays in the onset of exponential growth, slower growth rates, and limited maximum growth (Opicapone and Ebastine, respectively). Importantly, these compounds were structurally distinct from compounds in the training set, particularly antibiotics, further confirming our framework's ability to uncover structurally novel growth-inhibiting compounds. Screening additional strains, particularly more Gram-positive strains, could further uncover the growth-inhibitory activity of a greater number of predicted broad-spectrum antimicrobials[24]. While the compounds evaluated in this study require further research to be re-purposed into new antibiotic treatments, they are interesting study subjects when considering the effect of human-targeted drugs on microbial life. This is especially relevant, given the growing body of research showing that non-antibiotic drugs contribute to the appearance of bacterial strains resistant to current antibiotics[46–49]. Future work can investigate the molecular mechanisms of action of these compounds, leading to a better understanding of the effect non-antibiotic drugs have on microbial growth.

We consider our proposed framework as a first important step toward the more general goal of computationally-guided antimicrobial discovery in the face of data scarcity. We envision that several future research directions can improve the present framework. For example, the interpretability of MolE representation may be enhanced by enabling quick identification of molecular characteristics associated with properties of particular interest for microbiology. Our comparative analysis between MolE-based and ECFP4-based AP scores also suggests that ensemble scoring schemes that combine AP scores from multiple molecular representations, such as, e.g., MolFormer or ChemBERTa, may further improve prioritization accuracy. Likewise, alternative deep-learning architectures such as graph-attention networks or graph-transformer networks[50] may also improve downstream prediction tasks. Incorporating biological features of microbial strains into the representation learning process may also help improve the specificity of predictions, enabling the identification of effective narrow-spectrum treatments[4].

In conclusion, our proposed framework addresses key challenges in the field of antimicrobial discovery. By overcoming the need for large data resources, we envision that the presented workflow and methodologies such as ours are better poised to complement large screenings, thus increasing the rate at which new treatment candidates are uncovered.

## Methods

### Pre-training

Dataset: The MolE pre-training dataset was created by randomly sampling 100,000 unlabeled molecules from a collection of 10,000,000 unique structures originally collected by ChemBERTa[25]. This subset is then randomly split into training (90%) and validation (10%).

Molecular graph construction: Each molecule is represented as a graph, where all atoms are nodes $V$ and the chemical bonds between them are the edges $E$. The attributes encoded for each atom are the element it represents and its chirality type (tetrahedral CW, tetrahedral CCW, other). Likewise, each bond is embedded by its type (single, double, triple, aromatic), and its direction (none, end-upright, end-downright).

As shown in Fig. 1a, during pre-training two augmentations ($Y^A$ and $Y^B$) are created by following the subgraph removal procedure, first described in MolCLR[27]. Briefly, a seed atom is selected at random. This seed atom and its neighbors are masked and then neighbors of the neighbors are masked until 25% of the atoms present in the original have been masked. The bonds between these masked atoms are removed, producing a subgraph of the original molecule. This subgraph removal procedure is done for each individual augmentation. While this can result in very different subgraphs for the same compound entering the GNN backbone, we expect that by performing this procedure over several epochs and for several molecules, similar representations are learned for compounds with similar structures (Fig. 2).

Graph Neural Networks: We explored the ability of Graph Isomorphism Networks (GINs)[51] and Graph Convolutional Networks (GCNs)[11] to extract meaningful features from the molecular graphs constructed in the previous step. Both algorithms use an update function to learn an actualized representation of each node. In the case of GINs, this update is performed by an MLP head:

$$\mathbf{h}_v^{(l)} = MLP^{(l)}\left(\mathbf{h}_v^{(l-1)} + \sum_{u \in N(v)} \mathbf{h}_u^{(l-1)} + \mathbf{e}_{v,u}^{(l)}\right) \qquad (1)$$

where $\mathbf{h}_v^{(l)}$ is the updated vector representation of node $v \in V$ at the $l$-th GIN layer, $N(v)$ is the set of neighbors of $v$, and $\mathbf{e}_{v,u}$ represents the vector embedding of the attributes of the bond between $v$ and its neighbor $u$.

Molecular representation: To obtain a global vector representation of the molecular structure, we first gather a pooled, graph-level representation for each GNN layer ($\mathbf{g}^{(l)}$) by adding the node embeddings of all atoms in the molecule.

$$\mathbf{g}^{(l)} = \sum_{v \in V} \mathbf{h}_v^{(l)} \qquad (2)$$

A final graph representation $\mathbf{r}$ is obtained by concatenating the graph-level representations of each layer into a single vector.

$$\mathbf{r} = \text{CONCAT}\left(\mathbf{g}^{(l)} | l = 1, 2, ..., L\right) \qquad (3)$$

Here, $L$ represents the total number of GNN layers used, and CONCAT is the concatenation operator. In our setup, the dimensionality of $\mathbf{g}^{(l)}$ is set as a 200-dimensional vector. Given that $L = 5$, $\mathbf{r}$ is therefore a 1000-dimensional vector. The graph representation learned for augmentation $Y^A$ is correspondingly denoted as $\mathbf{r}^A$, as is the representation learned for augmentation $Y^B$ denoted as $\mathbf{r}^B$.

Non-contrastive learning: Once the final graph representations $\mathbf{r}^A$ and $\mathbf{r}^B$ are produced, an MLP layer is used to obtain embeddings $\mathbf{z}^A$ and $\mathbf{z}^B$, which are $D$-dimensional vectors. These embedding vectors are used to evaluate the $\mathcal{L}_{BT}$ objective function[30]. First, an empirical cross-

correlation matrix $\mathcal{C}$ is computed between $\mathbf{z}^A$ and $\mathbf{z}^B$:

$$\mathcal{C}_{ij} = \frac{\sum_b \mathbf{z}^A_{b,i} \mathbf{z}^B_{b,j}}{\sqrt{\sum_b \mathbf{z}^A_{b,i}} \sqrt{\sum_b \mathbf{z}^B_{b,j}}} \qquad (4)$$

Here, $b$ indexes the batch samples and $i$ and $j$ index the vector dimensions of the embeddings. In practical terms, $\mathcal{C}$ is a $D \times D$ matrix and represents the average cross-correlation matrix of a given batch $b$. Finally, $\mathcal{C}$ is used to calculate the $\mathcal{L}_{BT}$ loss:

$$\mathcal{L}_{BT} = \sum_i (1 - \mathcal{C}_{ii})^2 + \lambda \sum_i \sum_{i \neq j} \mathcal{C}_{ij}^2. \qquad (5)$$

In essence, by minimizing $\mathcal{L}_{BT}$ we minimize the difference between $\mathcal{C}$ and the identity matrix. In this way, embeddings obtained from augmentations of the same molecule are made to be invariant to distortions, while at the same time penalizing redundancy of information encoded by the components of the embedding. The positive constant $\lambda$ controls the trade-off between these two terms.

Pre-training setup: For all experiments, MolE pre-training is done over 1000 epochs with a batch size of 1000 molecules. An Adam optimizer with weight decay $10^{-5}$ is used to minimize $\mathcal{L}_{BT}$. The first 10 epochs are performed with a learning rate of $5 \times 10^{-4}$, after which cosine learning decay is performed. The parameter configuration that minimizes the validation error is kept for downstream tasks.

## Benchmarking
Datasets: We collected 11 benchmark datasets originally curated and made available by MoleculeNet[14]. This set included 6 binary classification tasks and 5 regression tasks. To make these benchmarking scenarios more challenging and realistic, we split each dataset into training (80%), validation (10%), and test (10%) sets following the scaffold splitting procedure described by Hu et al.[36]. Briefly, the Murcko scaffold of each molecule is determined, which identifies key topological landmarks in the overall structure[52]. Molecules that share the same scaffold are then collectively assigned to the same set. The aim of the scaffold splitting procedure is to create a realistic scenario where the molecules seen during training are structurally dissimilar to those seen during its application to a novel chemical library.

Training and evaluation: In our benchmarking experiments, MolE was pre-trained on 100,000 molecules from Pubchem[29]. Different dimensionalities of the embedding vectors ($\mathbf{z}^A$ and $\mathbf{z}^B$) and values for the trade-off parameter $\lambda$ were explored, while the dimensionality of the representation $\mathbf{r}$ is fixed as a 1000-dimensional vector. After pre-training, we evaluated the resulting representation in two ways: (i) the static vector representation $\mathbf{r}$ was used as input for machine-learning algorithms (termed MolE$_{\text{static}}$), or (ii) the weights learned by the GIN layers were used to fine-tune a predictor for a specific task (termed MolE$_{\text{finetune}}$).

In the first case, the pre-trained representation MolE$_{\text{static}}$ was used as input molecular features to Random Forest[35] or XGBoost[34] algorithms. Hyperparameter optimization is performed via random search (details in Supplementary Tables 4 and 5). Each model configuration was trained three times with different seed values. The model configuration with the largest mean ROC-AUC value on the validation set was then evaluated on the test set. For datasets with more than one task, the average performance per task is reported.

In the case of fine-tuning, an untrained MLP head is placed after the GNN layers. For all classification tasks, the Softmax cross-entropy loss is optimized, while in the case of regression, the $L_1$ loss is optimized for the QM7 and QM8 tasks and the mean squared error is optimized for all other tasks. A random search is performed for hyperparameter optimization (Supplementary Table 6). The selected architecture was trained for 100 epochs, using an Adam optimizer with

a weight decay of $10^{-6}$. While the GNN and the MLP are updated during training, the learning rate chosen for both parts differed. Each model configuration was trained three times.

Extended Connectivity Fingerprints (ECFP) were calculated using the functionality available in RDKit 2020.09.1.0[53]. In order to get ECFP4 fingerprints we set the relevant parameters fp_radius=2 and fp_bits=1024.

Other predictors: In Table 1 and Supplementary Table 1, performance metrics for GCN, GIN, SchNet, MGCN, D-MPNN, and Hu et al. are taken from the publication of MolCLR[27]. The ROC-AUC values for HiMol were taken from the respective publication[54], except for the HIV task which is evaluated in the current study.

The MolCLR$_{\text{GIN}}$[27] model made available on their GitHub page was used for all benchmarks, following the instructions in the same repository. The molecular representation obtained after GNN feature extraction was used as the input for either XGBoost or Random Forest. The N-Gram[55] and HiMol[54] models were pre-trained and molecular features were extracted following the default instructions and parameters made available in their respective GitHub repositories.

## Ablation study on the MolE framework
To identify the components of MolE that contribute to performance gains, we performed an ablation study shown in Supplementary Fig. 3.

GNN backbone and construction: We compared GINs and GCNs in our graph feature extraction step combined with two alternatives for constructing the molecular representation $\mathbf{r}$. Concatenated (C) representations are obtained as described previously (Eq. (3)). Non-concatenated (NC) representations consisted of the pooled output of the last GNN layer. In experiments where no concatenation is performed, $\mathbf{g}^{(l)}$ is set as a 1000-dimensional vector and the representation $\mathbf{r}$ remains a 1000-dimensional vector. We observed that models perform best when trained on representations built with the concatenated strategy, independent of the GNN backbone used during pre-training (Supplementary Fig. 3a). This indicates that each GNN layer captures important information about the molecular structure. In the ClinTox task, both GIN-derived representations outperformed their GCN counterparts.

Embedding dimensionality, trade-off parameter, and dataset size: We observed increased performance on the ClinTox task when $\mathbf{z}^A$ and $\mathbf{z}^B$ were 8000-dimensional vectors (Supplementary Fig. 3b). We also found performance increases as the $\lambda$ trade-off parameter approaches $10^{-4}$ (Supplementary Fig. 3c). Finally, we noted that the size of the unlabeled dataset used during pre-training does not necessarily improve performance on most classification tasks (Supplementary Fig. 3d). A higher ROC-AUC value can be observed in the ClinTox task when MolE's representation is pre-trained on 200,000 molecules.

The Barlow-Twins non-contrastive objective: Overall, we note that performance using MolCLR's representation learned from large-scale unlabeled data is on par with the performance obtained from our Barlow-Twins pre-training framework on our smaller set of unlabeled data. We see the largest performance improvement when concatenating the graph-level representation learned by each GIN layer. In our performance comparison, we denote the original representations as non-concatenating (NC) and the novel strategy as concatenating (C), respectively (see Supplementary Fig. 3e for details).

With these observations, we decided to pursue the task of antimicrobial discovery using the MolE$_{\text{static}}$ representation obtained after pre-training on 100,000 molecules with $\lambda = 10^{-4}$, $\mathbf{z}^{A,B} \in \mathbb{R}^{8000}$, using a GIN backbone and constructed by concatenating graph-layer representations.

## Exploring the MolE representation
Representation similarity: The distance between two MolE representations for compounds $i$ and $j$, $\mathbf{r}^i$ and $\mathbf{r}^j$, was determined using the

cosine distance:

$$d_{\text{cosine}}(\mathbf{r}^i, \mathbf{r}^j) = 1 - \frac{\mathbf{r}^i \cdot \mathbf{r}^j}{||\mathbf{r}^i||_2 ||\mathbf{r}^j||_2} \qquad (6)$$

The dissimilarity between two ECFP4 fingerprints, $\mathbf{r}^i_{\text{ECFP4}}$ and $\mathbf{r}^j_{\text{ECFP4}}$, was calculated using the Jaccard distance:

$$d_{\text{jaccard}}(\mathbf{r}^i_{\text{ECFP4}}, \mathbf{r}^j_{\text{ECFP4}}) = 1 - \frac{|\mathbf{r}^i_{\text{ECFP4}} \cup \mathbf{r}^j_{\text{ECFP4}}|}{|\mathbf{r}^i_{\text{ECFP4}}| + |\mathbf{r}^j_{\text{ECFP4}}|} \qquad (7)$$

Importantly, the Jaccard distance is equivalent to 1 - Tanimoto similarity (see Supplementary Fig. 2).

Uniform Manifold Approximation and Projection: A UMAP embedding based on the cosine distance between MolE representations was built using the umap-learn 0.5.3 Python module.

### Predicting antimicrobial activity on human gut bacteria

Dataset: The adjusted $p$ value table from Maier et al.[24] was used to determine labels for the growth-inhibitory effects of the screened compounds. Compound-microbe combinations with an adjusted $p$ value <0.05 were considered to be examples of growth inhibition. The 1197 molecules were divided into training (80%), validation (10%), and test (10%) sets following the scaffold splitting procedure.

Molecular and bacterial strain representation: We used the SMILES string of a given molecule $i$ to obtain the corresponding representation $\mathbf{m}_i$, which is a $d$-dimensional vector. In our work, $\mathbf{m}$ can be one of three possible representations: (i) ECFP4, in which case $d = 1024$, (ii) MolE$_{\text{static}}$ ($d = 1000$), and (iii) a set of explicit Chemical Descriptor features ($d = 98$), described by Algavi and Borenstein[4], which were calculated using RDKit.

A given microbial strain $j \in B$ (where $B$ is the complete set of bacterial strains) is represented as a one-hot-encoded vector $\mathbf{b}_j$. Given that 40 strains are present in the dataset, $\mathbf{b}_j$ is a 40-dimensional vector.

Compound-microbe predictions of antimicrobial activity: The compound and microbe vectors $\mathbf{m}_i$ and $\mathbf{b}_j$ are concatenated to form $x_{ij}$, which is a $40 + d$-dimensional vector. This combination of molecular and microbe representations is the input to our classifier function $f(\cdot)$. In our work, $f(\cdot)$ is an XGBoost model that, for each $x_{ij}$, outputs a probability ($p_{ij}$) that indicates the likelihood of compound $i$ inhibiting the growth of microbe $j$.

$$x_{ij} = \text{CONCAT}(\mathbf{m}_i, \mathbf{b}_j) \qquad (8)$$

$$p_{ij} = f(x_{ij}), \; p_{ij} \in [0, 1] \qquad (9)$$

Ranking compounds with Antimicrobial Potential scores: We summarize the predicted spectrum of activity of compound $i$ two ways:

(i) The total number of strains predicted to be inhibited by the compound $K_i$. This is achieved by thresholding the antimicrobial predictive probabilities ($p_{ij}$) into binary predictions of growth inhibition and adding up the number of strains predicted to be inhibited.

$$K_i = \sum_{j=1}^{40} \text{t}(p_{ij}), \; K_i \in [0, 1, 2, ..., 40] \qquad (10)$$

where

$$\text{t}(p_{ij}) = \begin{cases} 1, & p_{ij} \geq s \\ 0, & p_{ij} < s \end{cases} \qquad (11)$$

Here, the function $t(\cdot)$ binarizes the output by determining whether $p_{ij}$ exceeds a threshold $s$. We selected $s$ to optimize the F1-Score

metric obtained on the validation set for each model (Supplementary Fig. 14 and Supplementary Table 7). This optimized cutoff was 0.044, 0.068, and 0.209 for the model trained with MolE, Chemical Descriptors, and ECFP4 features, respectively.

(ii) We define the Antimicrobial Potential score $G_i$ for compound $i$ as the $\log_2$ of the geometric mean of the antimicrobial predictive probabilities $p_{ij}$ across all $j$ microbes:

$$G_i = \log_2 \left( \left( \prod_{j=1}^{40} p_{ij} \right)^{\frac{1}{40}} \right) \qquad (12)$$

Additionally, we consider the value of the Antimicrobial Potential score when calculated on the subsets of Gram-positive $G_i^+$ and Gram-negative $G_i^-$ microbes. In Eqs. (13), (14) we impose a fixed indexing on the set of taxa, where the first 22 indices represent all Gram-positive bacteria ($j \in [1, 2, \ldots, 22]$) and the remaining 18 indices represent all Gram-negative bacteria ($j \in [23, 24, \ldots, 40]$)

$$G_i^+ = \log_2 \left( \left( \prod_{j=1}^{22} p_{ij} \right)^{\frac{1}{22}} \right) \qquad (13)$$

$$G_i^- = \log_2 \left( \left( \prod_{j=23}^{40} p_{ij} \right)^{\frac{1}{18}} \right) \qquad (14)$$

Model selection and evaluation: A random search over XGBoost hyperparameters was performed for each chemical representation. The model configuration with the highest PR-AUC on the validation set was then evaluated on the test set.

### Predicting antimicrobial compounds in an orthogonal chemical library

Chemical library: A separate chemical library was constructed based on the FDA-approved Drugs, Human Endogenous Metabolite, and Food-Homology Compound libraries made available by MedChemExpress https://www.medchemexpress.com/. The chemical structures for these compounds were gathered from PubChem[29] using the pubchempy 1.0.4 Python module. SMILES were canonicalized and salts were removed using RDKit[53].

Compound annotation: Information provided by MedChemExpress included descriptions of the chemicals in the library. The corresponding Anatomical-Therapeutic-Chemical (ATC) code was assigned to each compound by matching compound name strings. A complete collection of ATC codes was gathered from https://github.com/fabkury/atcd. Overlap with chemicals in the library used by Maier et al.[24] was determined by matching chemical names and ATC codes. Chemicals present in both libraries were not considered for downstream prediction. SMILES were gathered from PubChem using the Python module We also use the pubchempy v1.04.

Prediction and evaluation: Molecules with $K_i \geq 10$ were prioritized for further evaluation. We performed a literature search for articles available on PubMed[56] that described the in-vitro and/or in-vivo antimicrobial activity of our prioritized compounds against any bacterial species. When recording MICs, we considered the lowest concentration at which no growth was observed for any Gram-positive or Gram-negative strain.

### Experimental validation

Compound prioritization: In total 6 compounds were selected for experimental validation. Criteria considered for compound selection were the following: (i) The compound was predicted to inhibit 10 strains or more, (ii) the compound was not an antibiotic, (iii) the compound did not have antifungal activity, (iv) no previous literature

describing the antimicrobial activity of the compound was found, and (v) the compound could be purchased through an independent provider. Furthermore, we attempted to choose compounds with different biological functions that were structurally diverse from each other. Additional information about the chosen compounds can be found in Supplementary Table 8.

Bacterial strains and growth conditions: Before the experiments, all strains were cultured overnight in Lysogeny Broth (LB Lennox) adjusted to pH 7.5 at 37 °C. Detailed information about the strains used in this study can be found in Supplementary Table 9.

Measurement of Minimum Inhibitory Concentrations: All compounds were purchased from Biomol GmbH (Germany). Stock solutions were prepared in DMSO and stored at − 20 °C until further use. MIC measurements were performed in 96-well plates with 100 $\mu$L bacterial cultures in Mueller Hinton (MH) broth using 1:2 serial dilutions of the tested compounds. Starting concentrations of 128 $\mu$g/mL were used for Cetrorelix, Opicapone, Thymidine, Visomitin, and Elvitegravir, and 64 $\mu$g/mL for Ebastine due to low solubility. No-compound controls contained DMSO or Water. Overnight cultures of three biological replicates of each bacterial strain were adjusted to $OD_{600} = 0.1$ and inoculated into the plates by pinning using a Singer Rotor (Singer Instruments, UK), achieving a 1:200 dilution. Plates were sealed with transparent breathable membranes (Breathe-Easy®, Sigma-Aldrich-Merck) and incubated at 37°C in a Cytomat 2 incubator (Thermo Scientific) with continuous shaking at 800 rpm. $OD_{600}$ was measured at regular 30-minute intervals for up to 12 h in a Synergy H1 plate reader (Agilent, USA). Additional information about the tested bacterial strains can be seen in Supplementary Table 8.

Growth curve modeling: All growth curves were normalized by subtracting the minimum of the second, third, and fourth measurements taken. Afterward, each individual curve was modeled as a logistic curve with `Sicegar` 0.2.4 R package. From these curves, parameters such as the maximum growth rate ($\mu_{max}$), end of lag-phase, start of stationary phase, and the carrying capacity were extracted. The maximum doubling time $t_d$ was estimated as:

$$t_d = \frac{\ln 2}{\mu_{max}} \quad (15)$$

### Software
The MolE pre-training framework was implemented using the pytorch-geometric 1.6.3 framework[57] with Python 3.7[58]. We use the `RandomForestClassifier` and `RandomForestRegressor` implementation available in scikit-learn 1.0.2[59] and the XGBClassifier and XGBRegressor objects from xgboost 1.6.2[60]. The scikit-learn 1.0.2[59] module is also used when computing ROC-AUC, PR-AUC, and F1 score metrics. The R 4.3.1 language was used for its ggplot2 3.4.2 for plotting, Sicegar 0.2.4 packages. ECFP4, chemical descriptors, and general SMILES processing were done with the rdkit 2020.09.1.0 package.

### Reporting summary
Further information on research design is available in the Nature Portfolio Reporting Summary linked to this article.

## Data availability
The unlabeled chemical structures used for pre-training were gathered from the MolCLR GitHub repository https://github.com/yuyangw/MolCLR. The adjusted p-value table from[24] was used to train models to predict antimicrobial activity and was gathered from the respective publication. The chemical library from MedChemExpress can be found in our GitHub repository https://github.com/rolayoalarcon/mole_antimicrobial_potential[61]. Results for predicting antimicrobial activity, and the data from experimental validation are available at https://github.com/rolayoalarcon/mole_antimicrobial_potential[61]. All data is publicly available and can be accessed without restrictions. Source data are provided with this paper.

## Code availability
The code for the MolE pre-training framework can be found at https://github.com/rolayoalarcon/MolE[62]. Code for predicting antimicrobial activity can be found at https://github.com/rolayoalarcon/mole_antimicrobial_potential[61]. The pre-trained model used for the prediction of antimicrobial potential is available in Zenodo https://doi.org/10.5281/zenodo.10803099[63].

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

## Acknowledgements

This work was funded by a grant awarded to C.L.M., A.R.B and C.M.S. for the StressRegNet consortium within the Bavarian research network bayresq.net funded through the Bavarian State Ministry of Science and Arts, Germany.

## Author contributions

R.O.A. and C.L.M. conceived the overall objectives and design of the project. R.O.A., C.L.M. and M.R. contributed to the development and

evaluation of the MolE pre-training framework. R.O.A., A.R.B., C.M.S. A.Z. and M.K.A. conceived the experimental validation of antimicrobial activity for selected compounds. A.Z. and M.K.A. performed validation experiments of antimicrobial activity. R.O.A. and M.B. analyzed data from experimental validations. R.O.A. implemented all computational methods. All authors revised and approved the final version of the Article.

## Funding

## Competing interests
The authors declare no competing interests.
