## [Transparent Peer Review file · Nature Communications]

Pre-trained molecular representations enable antimicrobial discovery

Corresponding Author: Mr Roberto Olayo-Alarcon

Version 0:

Reviewer comments:

Reviewer #1

(Remarks to the Author)

The authors introduce MolE, a method for molecular representation learning that leverages unlabeled chemical structure information to derive a more general representation of molecules. Using MolE, they further design a predictive model that assesses compounds for their antimicrobial potential. The model was experimentally validated, and three out of five tested compounds were confirmed to have measurable effects on the growth of the Gram-positive pathogen *S. aureus*.

Advantages:

1. The work proposes a novel protocol for ligand-based broad-spectrum antibacterial discovery.
2. The method has been validated through wet-lab experiments, providing a good example of combining computational and experimental approaches.

Disadvantages:

1. The computational method lacks novelty. As a computationally focused paper, my main concern is the novelty of the methodology. The molecular representation part of this method is very similar to MolCLR, and might even be considered a degenerate version. Although the authors introduce the Barlow-Twins pre-training loss, the motivation is unclear, and without specific ablation experiments, I cannot determine how the non-contrastive Barlow-Twins pre-training framework contributes to the model's performance. Considering the rapid advancements in SSL methods the authors need to clarify the novelty and advancement of their methodology.
2. The claims are not well substantiated. The authors claim their method is competitive, but compared to the results reported in MolCLR's fine-tuning version, their method does not show significant advantages. For instance, MolCLR reports a Clitox result of 93.2, whereas this paper reports 87.7. The authors need to clarify why their results are lower than those reported in other papers. Currently, I can't agree with the authors' claims.
3. The experimental validation is not solid enough. The authors find that less than 6.8% of molecules in a commercial library are non-antibiotics and try to prove that many of them are, in fact, antibiotics. This protocol is odd because the fine-tuning library also includes many FDA-approved drugs, there may be data leakage. Moreover, this scenario does not align well with real-world applications. In antibiotic discovery, we prefer to find new series of antibiotics from a large, novel compound library (rather than repurposing) to address resistance issues. The authors should demonstrate their method's predictive ability on new molecules.

Additional Questions:

1. In the Method section and Figure 1: "Briefly, a seed atom is selected at random. This seed atom and its neighbors are masked, and then neighbors of the neighbors are masked until 25% of the atoms present in the original have been masked." Is the subgraph removal procedure performed twice randomly? Does this process consider the impact of different initial nodes on the representation? For example, the representation of the graph with key nodes removed may change significantly.
2. In "MolE enables competitive molecular property prediction": "Finally, the benchmarks also revealed MolE's non-contrastive Barlow-Twins strategy to be superior to the state-of-the-art contrastive MolCLR framework." Have you tried using the contrastive learning method and compared it with the "Non-contrastive learning" approach in the same model?
3. Is there any overlap between the 2,327 molecules in the chemical library and the 1,197 marketed drugs for finetuning? Both contain FDA-approved drugs.
4. From the results in Table 1, it seems that the performance of various methods across different datasets varies greatly. What

accounts for this disparity? Could you provide more details and explore which types of datasets MoIE demonstrates optimal suitability?

5. The authors should consider providing more details on the experimental setup of "Experimental Validation," including the rationale for the choice of bacterial strains and concentrations used in the assays.

6. In Figure 3d, it is interesting that MoIE shows good performance in the "hard-to-recall sample range" (PRC, Precision < 0.75). Could you provide the overlap of hard-to-recall samples in MoIE, ECFP4, Chem. Descriptors, etc.?

(Remarks on code availability)

The code is partially reproducible and the README file is clear enough.

Reviewer #2

(Remarks to the Author)

(Remarks on code availability)

Reviewer #3

(Remarks to the Author)

Summary

Olayo-Alarcon et al describe a representation learning model for antibiotic prediction tasks. Their approach, named MoIE (molecular representation though redundancy reduced embedding), learns chemical representations in a task-independent manner using GINs. These learned representations can then be used for downstream prediction tasks. In this study, MoIE representations are used for the classification of molecules as antibacterial against a panel of 40 human commensal bacterial species. The authors' model (MoIE-XGBoost) was trained on a collection of 1,197 compounds tested against 40 representative commensal isolates. This model was then applied to a separate library of 2,327 bioactive molecules to predict molecules with antibacterial activity. Five molecules were selected for in vitro experimentation against four pathogen bacterial species (three Gram-negative and one Gram-positive). Two molecules displayed MICs against at least one bacterium.

Code

1. The code is well-constructed. Code was easily accessible and worked well off the shelf.

2. The authors are encouraged to review the listed dependencies (for example, Pubchempy, pyyaml, matplotlib, seaborn and openpyxl were not included but required by the software).

3. Minor issues were encountered when running the prepared Jupyter notebooks.

4. Both model and fine-tuning scripts currently read a specific yaml file for its parameters; I suggest making the model more convenient to customize by users by integrating command-line arguments to manually specify training parameters.

Comments

1. The paper is straightforward and concise. Very well written.

2. Figure 2. How does a UMAP generated using ECFP4 fingerprints visually compare to the UMAP shown in Figure 2a? This would be a useful visual comparison to show the ability of MoIE to generate more ideal molecular representations relative to ECFP4, which is a common method to generate representations.

3. Figure 2. What are the Tanimoto nearest neighbours in the test set to the query molecule in Figure 2c? How do these Tanimoto nearest neighbours compare to those identified by MoIE representations and ECFP4 representations?

4. Figure 3. It would be useful to include a visual representation of the agreement (or lack thereof) of predictions when using chemical descriptors, ECFP4 representations, and MoIE representations. Figure 3e begins to show this for a single molecule, but it would be much more informative to summarize how the different models' predictions (using different representations) compare. On which molecules do the models agree? On which molecules do the models disagree? Are different representations better/worse in different regions of chemical space?

5. Figure 4. How does the chemical space of the 2,327-compound library compare to the chemical space of the 1,197-compound library?

6. There should be more elaboration in the main text on how "antimicrobial potential" is calculated and used to guide downstream molecule prioritization.

7. The MoIE-XGBoost model is applied to the 2,327-compound library (Figure 4a), and five antibacterial predictions are tested in the laboratory against three Gram-negative pathogens and one Gram-positive pathogen. However, this model is

trained using 40 human commensal strains. The authors need to discuss in greater detail the phylogenetic relationship between these four pathogens and the commensal isolates from the training dataset.

8. The antibacterial efficacies of opicapone and ebastine are minimal and, in my opinion, should not be considered antibacterial.

9. Figure 4 and Figure 5 – it would be a tremendously useful addition to the paper to perform the same set of experiments (predictions and in vitro validations) using a model trained using ECFP4 representations as a common method to vectorize molecules. This would serve as the ideal comparison of MoIE representations in the context of wet lab validation of model predictions using different molecular representations.

10. Discussion – there is a reference to “Supplementary Table ??”.

(Remarks on code availability)

Version 1:

Reviewer comments:

Reviewer #1

(Remarks to the Author)

Thank you to the authors for their detailed responses and the additional experiments. I have also reviewed the questions and replies from other reviewers, and I believe my main concerns have been well addressed. Good luck.

(Remarks on code availability)

Reviewer #3

(Remarks to the Author)

The manuscript is improved and I thank the authors for their efforts. There are potential benefits to MoIE representations over conventional ECFP4 fingerprints, although much room for improvement exists.

My remaining major concern is in the experimental validation of model predictions (Figure 5). Neither opicapone or ebastine would be classified as a reasonable antibiotic given the lack of an observable MIC at any concentration shown in the manuscript. I understand that growth is suppressed, but at no concentration is complete growth inhibition observed. This does, unfortunately, limit the impact of the study.

The authors need to provide detailed explanations as to why more potent antibiotics (at least molecules that have an MIC) weren't identified. Moreover, an obvious question that could be addressed is, would another ML method (perhaps not using MoIE representations) predict more potent molecules (defined by in vitro MIC) from this same chemical set? This ultimately defines the true utility of a molecular property predictor.

(Remarks on code availability)

No issues encountered.

Version 2:

Reviewer comments:

Reviewer #3

(Remarks to the Author)

The authors have addressed my remaining comments. I appreciate the work that went into the paper.

(Remarks on code availability)

Response letter to reviewer comments for Manuscript

NCOMMS-24-40179A "Pre-trained molecular representations enable antimicrobial discovery"

We thank the reviewers and the editor for their time and their valuable feedback regarding the present manuscript. To the best of our ability, we have taken all reviewer comments into account and adapted the work accordingly.

We performed new computational ablation studies, compared different models, created new Figures (both for the response letter and the manuscript), and adapted the text in the manuscript.

We also ensured to address the specific concerns of Reviewer #1 relating to the strength of experimental validation and data leakage (which is NOT present), as well as adapted language usage relating to antibiotic/antimicrobial activity.

REVIEWER COMMENTS

Reviewer #1 (Remarks to Author):

The authors introduce MoIE, a method for molecular representation learning that leverages unlabeled chemical structure information to derive a more general representation of molecules. Using MoIE, they further design a predictive model that assesses compounds for their antimicrobial potential. The model was experimentally validated, and three out of five tested compounds were confirmed to have measurable effects on the growth of the Gram-positive pathogen *S. aureus*

Advantages:

- 1.The work proposes a novel protocol for ligand-based broad-spectrum antibacterial discovery.
- 2.The method has been validated through wet-lab experiments, providing a good example of combining computational and experimental approaches.

We thank the reviewer for their recognition of the value of our work.

Disadvantages:

1.The computational method lacks novelty. As a computationally focused paper, my main concern is the novelty of the methodology. The molecular representation part of this method is very similar to MolCLR, and might even be considered a degenerate version. Although the authors introduce the Barlow-Twins pre-training loss, the motivation is unclear, and without specific ablation experiments, I cannot determine how the non-contrastive Barlow-Twins pre-training framework contributes to the model's performance. Considering the rapid advancements in SSL methods the authors need to clarify the novelty and advancement of their methodology.

We thank the reviewer for this comment. While it is true that most of the individual components of our pre-training framework have been described before, to our knowledge, we are the first to combine the use of Barlow-Twins to learn an unsupervised graph representation of *molecular structures*. Prior research only used it for graphs arising in social network research. We chose the Barlow-Twins framework because of its reported superior transfer learning capabilities and insensitivity to hyper-parameter choices when compared to its contrastive counterparts [30]. We added a sentence to the main manuscript to make the motivation clear (see page 3, l. 89-90 in the revised manuscript). More generally, our main emphasis was indeed to present a light-weight, fully reproducible and transparent end-to-end pipeline that (i) learns from scratch an unsupervised representation of molecules from unlabeled data, (ii) uses labeled data to design a generalizable molecule prioritization scheme (our AP scores), (iii) makes an informed selection of molecules based on these scores on an unseen molecule library, and (iv) tests the molecules in the lab for antimicrobial activity. We give further info when responding to Additional Question #2 from Reviewer #1.

2. The claims are not well substantiated. The authors claim their method is competitive, but compared to the results reported in MolCLR's fine-tuning version, their method does not show significant advantages. For instance, MolCLR reports a Clitox result of 93.2, whereas this paper reports 87.7. The authors need to clarify why their results are lower than those reported in other papers. Currently, I can't agree with the authors' claims.

We thank the reviewer for this observation. The metrics reported in our manuscript for the finetuned version of MolCLR are a result of our attempt to reproduce the results, using the the code and pre-trained model provided in the MolCLR repository (<https://github.com/yuyangw/MolCLR>, commit hash: fe603e04807eab7a6c1e02e28881f72e2c67bcde). One reason for the difference in performance metrics can be due to differences in our fine-tuning strategies. In the MolCLR paper (<https://www.nature.com/articles/s42256-022-00447-x>), the authors state in Supplementary Table 6 (https://static-content.springer.com/esm/art%3A10.1038%2Fs42256-022-00447-x/MediaObjects/42256_2022_447_MOESM1_ESM.pdf) "In addition, we randomly pick MolCLR-trained GNNs at different epoch as the initialization for fine-tuning." In our work, we always use the GNN weights that minimized our pre-training objective. This is also what the authors of MolCLR provide in their repository. Also, we always use the MolCLR_{GIN} model in our benchmarks. Other papers cited in this study, such as HiMol (<https://www.nature.com/articles/s42004-023-00825-5>), also report a lower ROC-AUC metric for the ClinTox task for MolCLR (90.4, <https://www.nature.com/articles/s42004-023-00825-5/tables/1>). Further details about our benchmarking can be found in the Methods section of our manuscript, under the subsection Benchmarking - *Other predictors*.

3. The experimental validation is not solid enough. The authors find that less than 6.8% of molecules in a commercial library are non-antibiotics and try to prove that many of them are, in fact, antibiotics. This protocol is odd because the fine-tuning library also includes many FDA-approved drugs, there may be data leakage. Moreover, this scenario does not align well with real-world applications. In antibiotic discovery, we prefer to find new series of antibiotics from a large, novel compound library (rather than repurposing) to address resistance issues. The authors should demonstrate their method's predictive ability on new molecules.

We thank the reviewer for raising this important point. It is true that the library used by Maier, et. al. and the orthogonal chemical library obtained from MedChemExpress (MCE) both contain FDA-Approved drugs. However, this is a very broad categorization, and there is a large chemical diversity within. In our work, we took measures to ensure that any molecules in the MCE library that were also present in the Maier et.al. library were **removed** and not considered for any downstream prediction, analysis, or considered for experimental validation. Therefore, we are sure that none of the non-antibiotic drugs predicted to have broad-spectrum activity are present in the training dataset and have thus not been "seen" by our model before. A more detailed description of our efforts to avoid data leakage can be found in Methods, subsection "Predicting antimicrobial compounds in an orthogonal chemical library". The code used to remove overlapping compounds can be found in our mole_antimicrobial_prediction repository. (https://github.com/rolayoalarcon/mole_antimicrobial_potential/blob/main/workflow/04.new_predictions.ipynb).

The reviewer correctly points out that structurally novel antibiotics are desired. Recent examples of such molecules, Halicin (<https://pubmed.ncbi.nlm.nih.gov/32084340/>) and Abaucin (<https://www.nature.com/articles/s41589-023-01349-8>), were discovered by making predictions on the Drug Repurposing Hub (<https://www.broadinstitute.org/drug-repurposing-hub>). Therefore, drug repurposing has been shown to be a valid strategy for uncovering structurally novel antibiotic candidates. In our work, we show that our Mole-XGBoost model is able to recover the known broad-spectrum activity of Halicin, thereby showing that our model can recognize the kind of molecules that are desired.

To further address the concerns of the Reviewer, we provide a quantitative analysis where we calculate the **molecular similarity** between the **compounds chosen for experimental validation** and the compounds in the Maier et.al. library used for model training. Recent literature dealing with the prediction of antimicrobial compounds suggests a Tanimoto similarity (=1-Jaccard distance) threshold between 0.3 and 0.4 is a good way of determining that two compounds are structurally dissimilar (<https://pubmed.ncbi.nlm.nih.gov/32084340/>, <https://www.nature.com/articles/s41589-023-01349-8>, <https://www.nature.com/articles/s41467-023-39264-0>). Using a Tanimoto similarity threshold of 0.3 we show below that all of the **compounds** chosen for **experimental validation** are **structurally dissimilar** to the antibiotics present in the training set. Indeed, the compounds with the strongest effect on the growth of *S. aureus*, Elvitegravir and Opicapone, are structurally dissimilar to the **entire** training set (!). The same is true for our positive control, Visomitin. Ebastine shares a higher degree of

similarity with a few compounds in the training set, including non-antibiotic compounds with broad-spectrum activity. These results show that our model can recover growth-inhibiting compounds from a novel chemical library with structurally diverse compounds it has not “seen” before.

Tanimoto similarity between experimentally validated compounds and compounds from Maier, et. al. used for model training. The dashed line indicates a Tanimoto similarity of 0.3. The most similar compound overall, most similar antibiotic, and the most similar non-antibiotic compound with broad-spectrum activity (BSA) are highlighted with the respective Tanimoto similarity. In the case of Elvitegravir, the most similar compound is an antibiotic. For Opicapone and Thymidine, the most similar compound is a non-antibiotic with BSA.

To further visually support our claims, we provide a joint UMAP embedding of the chemical space covered by the MCE library and by the Maier, et. al. library (see response to Reviewer #3 question 5). Here, we also see that the MCE library covers a considerably “larger space”. Even in this extended chemical space, our model is able to recognize the activity of compounds that are different from those present in the Maier, et. al. dataset.

We have added the figure above to the supplementary material as Supplementary Figure 9. We have also included a sentence in the main text of our manuscript, highlighting the result from this analysis (see page 10, l. 256-258):

“Importantly, the chosen compounds were all structurally distinct from the antibiotics in our training set (Tanimoto similarity \leq 0.3). Furthermore, Cetorelix, Elvitegravir, Opicapone, and Visomitin shared low molecular similarity to all compounds seen during model training (Supplementary Figure 9).”

Finally, as a point of clarification, we use MolE’s **static** representation as input for our XGBoost (or Random Forest) model which makes predictions of antimicrobial activity. No finetuning on the representation part is done for this task.

Additional Questions:

1. In the Method section and Figure 1: “Briefly, a seed atom is selected at random. This seed atom and its neighbors are masked, and then neighbors of the neighbors are masked until 25% of the atoms present in the original have been masked.” Is the subgraph removal procedure performed twice randomly? Does this process consider the impact of different initial nodes on the representation? For example, the representation of the graph with key nodes removed may change significantly.

We thank the reviewer for their comment. We realize that we should have included this detail in the description of our method (we added a description now). Indeed, the subgraph removal procedure is performed twice, each time the seed node is chosen randomly. We expect that, even if this procedure results in very different subgraphs entering our representation learning framework, the fact that we perform these steps over several epochs, for several molecules, will eventually lead to similar structures having similar representations. This is confirmed by our results in Figure 2.

We have added the following clarifying sentence on page 13, l. 357-260:

“This subgraph removal procedure is done for each individual augmentation. While this can result in very different subgraphs for the same compound entering the GNN backbone, we expect that by performing this procedure over several epochs and for several molecules, similar representations are learned for compounds with similar structures (Figure 2).”

2. In "MoE enables competitive molecular property prediction": "Finally, the benchmarks also revealed MoE's non-contrastive Barlow-Twins strategy to be superior to the state-of-the-art contrastive MolCLR framework." Have you tried using the contrastive learning method and compared it with the "Non-contrastive learning" approach in the same model?

We thank the reviewer for this suggestion. We did a performance comparison (and small ablation study) to address this question, summarized in the figure below (and added to the Supplementary Material).

We considered the following setup. The results in the original MolCLR publication [27], (e.g., for the HIV benchmark), indicate that MolCLR's performance considerably improves with the number of unlabeled data. We thus used the published large-scale pretrained MolCLR representation as baseline and trained XGBoost and Random Forest models on that representation. The results are shown in the Figure below as last entry in each panel.

We sought to evaluate how well MoE fares with the **Barlow-Twin objective** against this baseline, given that our framework is intended for smaller unlabeled data sets (e.g., 100,000 compounds). We compared three different settings for MoE architectures. Graph-Convolutional Networks (GCN) and Graph-Isomorphism Networks (GIN) with their final representation being the last graph layer (so called non-concatenating (NC) representation) and our proposal of concatenating all layers into one final representation (so called concatenating (C) strategy). For all combinations, we test the performance of XGBoost and Random Forests algorithms (with **identical parameters**) on the selected benchmark tasks, highlighting the default strategy (GIN-C) as the first entries in each panel below.

Comparison of pre-training objectives. The performance of MolCLR is compared to that of equivalent architectures trained with the Barlow-Twins objective. GCN and GIN architectures are evaluated without concatenating the learned representations of each layer (NC). Our final architecture using GIN and concatenating the representation of each GNN layer is shown for reference (GIN-C*)

Overall, we note that MoIE (with smaller pretraining sets) is largely on par with the large-scale MolCLR framework (on par except for ClinTox). We also observe that the concatenating strategy boosts the performance of standard GINs considerably and outperforms the large-scale MolCLR framework. This suggests that the C-strategy may be worth investigating also with respect to MolCLR and other frameworks. This will be subject to future research.

We have included the figure above in Supplementary Figure 3e. We have also updated our discussion of the benchmarking results to reflect the observations mentioned previously. On page 6, l. 159-161:

“Finally, the benchmarks and the ablation study also revealed MoIE’s pre-training architecture, in combination with the non-contrastive Barlow-Twins strategy, to be superior to the state-of-the-art contrastive MolCLR framework (Table 1 and Supplementary Figure 3e).”

3. Is there any overlap between the 2,327 molecules in the chemical library and the 1,197 marketed drugs for finetuning? Both contain FDA-approved drugs.

This was also a very important issue to us at the beginning of this project, and as detailed above, we made sure no overlap exists between the Maier, et. al. and MCE-derived libraries. The MCE-derived library is truly a new discovery set.

4. From the results in Table 1, it seems that the performance of various methods across different datasets varies greatly. What accounts for this disparity? Could you provide more details and explore which types of datasets MoIE demonstrates optimal suitability?

We thank the reviewer for this observation. To be frank, we do not have a proper explanation for the behavior of the different methods, including MoIE, on the MoleculeNet benchmarks. We observe that MoIE mirrors the performance of the other methods in terms of absolute performance. When taking into account the performance on the regression tasks, we hypothesize that MoIE is expected to perform well for datasets with fewer training examples (say, 1K-4K) and fewer overall objectives.

5. The authors should consider providing more details on the experimental setup of "Experimental Validation," including the rationale for the choice of bacterial strains and concentrations used in the assays.

We thank the reviewer for this suggestion. A similar request was made by Reviewer #3, we've added text in our manuscript, providing more details (see response for location in the revised manuscript). Briefly, Bacterial strains were selected to cover a wide range of Gram-negative and Gram-positive pathogens of concern - most of them part of the ESKAPE list (<https://www.dzif.de/en/glossary/eskape>). The selection was not based on taxonomy, but rather on potential of application. We decided to include a commensal *E. coli* strain as a representative of a member of the Human gut microbiome for reference. The cultivation method chosen for measurement of the minimum inhibitory concentration, as well as the concentration ranges, are in line with well-established and highly standardized methods for determination of antimicrobial activity.

6. In Figure 3d, it is interesting that MoIE shows good performance in the "hard-to-recall sample range" (PRC, Precision < 0.75). Could you provide the overlap of hard-to-recall samples in MoIE, ECFP4, Chem. Descriptors, etc.?

We thank the reviewer for pointing this out. Below, we show the intersection of positive samples present in the hard-to-recall range (precision ≤ 0.75) for our three models. Overall, similar samples are present in this range for all models, with MoIE having a lower number of samples in this range overall (panel a). The same is also true for true samples involving compounds with broad-spectrum activity (BSA) (panel b). It also seems that similar BSA compounds are present in this range.

Overlap of hard-to-recall samples between models. **a.** All true compound-microbe growth inhibition samples in the hard-to-recall range. **b.** True samples in the hard-to-recall range, for compounds with experimentally determined broad-spectrum activity (BSA). **c.** Broad-spectrum compounds with at least one sample in the hard-to-recall range.

Reviewer #1 (Remarks on code availability):

The code is partially reproducible and the README file is clear enough.

We thank the reviewer for their time in looking at our code. We have made several changes that should considerably improve reproducibility.

Reviewer #2 (Remarks to the Author):

We thank the reviewer for their time and willingness to participate!

Reviewer #3 (Remarks to the Author):

Summary

Olayo-Alarcon et al describe a representation learning model for antibiotic prediction tasks. Their approach, named MoIE (molecular representation though redundancy reduced embedding), learns chemical representations in a task-independent manner using GINs. These learned representations can then be used for downstream prediction tasks. In this study, MoIE representations are used for the classification of molecules as antibacterial against a panel of 40 human commensal bacterial species. The authors' model (MoIE-XGBoost) was trained on a collection of 1,197 compounds tested against 40 representative commensal isolates. This model was then applied to a separate library of 2,327 bioactive molecules to predict molecules with antibacterial activity. Five molecules were selected for in vitro experimentation against four pathogen bacterial species (three Gram-negative and one Gram-positive). Two molecules displayed MICs against at least one bacterium.

Code

1.The code is well-constructed. Code was easily accessible and worked well off the shelf.

We thank the reviewer for thoroughly revising our code. We have made several changes, including the ones suggested below.

2.The authors are encouraged to review the listed dependencies (for example, Pubchempy, pyyaml, matplotlib, seaborn and openpyxl were not included but required by the software).

We thank the reviewer for bringing this to our attention. We have listed all of the necessary in an environment.yaml file which can also be used to set up a virtual environment. More detailed instructions can be found in the README of our repositories.

3.Minor issues were encountered when running the prepared Jupyter notebooks.

We hope that by listing all of the necessary dependencies, all notebooks can be run without any issues.

4.Both model and fine-tuning scripts currently read a specific yaml file for its parameters; I suggest making the model more convenient to customize by users by integrating command-line arguments to manually specify training parameters.

We thank the reviewer for this suggestion. We have now updated all pre-training, finetuning, ML model training, and representation extraction scripts in such a way that they can accept parameters in a yaml file **and** as command line arguments. We hope that this facilitates the use of our code. Parameter documentation, as well as examples of how to use our scripts, are present in our READMEs. Any further suggestions are welcome (also via GitHub Issues, of course).

Comments

1. The paper is straightforward and concise. Very well written.

Thank you!

2. Figure 2. How does a UMAP generated using ECFP4 fingerprints visually compare to the UMAP shown in Figure 2a? This would be a useful visual comparison to show the ability of MoIE to generate more ideal molecular representations relative to ECFP4, which is a common method to generate representations.

We thank the reviewer for the suggestion. We show the UMAP embedding of the same collection of 100K molecular structures obtained after representing them with MoIE and ECFP4 representations. The same UMAP hyperparameters are used, with the exception of distance ('jaccard' is used for ECFP4 and 'cosine' is used for MoIE). Overall, we notice that the ECFP4 representation UMAP has less structure than MoIE's. Without overinterpreting UMAPs too much (we are aware of the criticisms in the comp. bio. community), this high-level visual analysis suggests that ECFP4-based representations provide less crisp differences between groups of chemically distinct molecules. To make this general observation a bit more concrete, we place the same highlighted structures as in the original MoIE representation for comparison, such as, e.g., amino acids or long Naphthalenes.

Comparing MoIE and ECFP4 representations of the same collection of 100,000 compounds from PubChem. The location of the same molecules is highlighted in both representations.

We have included the ECFP4 panel from the figure above as Supplementary Figure 1.

2. Figure 2. What are the Tanimoto nearest neighbours in the test set to the query molecule in Figure 2c? How do these Tanimoto nearest neighbours compare to those identified by MoIE representations and ECFP4 representations?

We thank the reviewer for this question. When comparing ECFP4 representations, we relied on the Jaccard distance (J). In our context, this distance is actually related to the Tanimoto similarity (which is the term more commonly used in the cheminformatics literature) via $Jaccard = 1 - Tanimoto$ (see, e.g., here <https://bmcbioinformatics.biomedcentral.com/articles/10.1186/s12859-019-3118-5>). To illustrate we made the figure below and included it as Supplementary Figure 2.

Molecular similarity as determined by MolE and ECFP4. Cosine distance is used to compare MolE representations, while the Jaccard (J) distance and Tanimoto (T) similarity are used for ECFP4. **a.** Molecular similarity ranking with Ractopamine (PubChem ID: 56052) as query. Top 4 most similar molecules according to MolE and ECFP4 (right). Distance with respect to all other molecules in the search space (left). **b.** Molecular similarity ranking with PubChem ID: 12277389 as query. **c.** Molecular similarity ranking with PubChem ID: 98701517 as query.

4. Figure 3. It would be useful to include a visual representation of the agreement (or lack thereof) of predictions when using chemical descriptors, ECFP4 representations, and MoIE representations. Figure 3e begins to show this for a single molecule, but it would be much more informative to summarize how the different models' predictions (using different representations) compare. On which molecules do the models agree? On which molecules do the models disagree? Are different representations better/worse in different regions of chemical space?

We thank the reviewer for this very interesting question. We have performed a deeper analysis on the models' predictions for the test set molecules described in Figure 3. Below, we compare the predicted number of bacterial strains inhibited of each model against the experimentally determined ground truth (last row). Overall, all models are able to recover the activity of compounds whose intended use is targeted at bacteria and fungi. The differences between models can be observed in their predictions for human-targeted drugs. The model using Chemical Descriptors makes many false positive predictions, while the ECFP4 model is often unable to recognize any activity for these compounds. A balance between these two behaviors is accomplished when using MoIE's pre-trained representations. We added this Figure as Sub-Figure (Fig. 3g) to the main manuscript.

Comparison of predicted broad-spectrum activity for compounds in the test-set. Each column is a compound, and they are grouped by their intended target species (Maier L., et. al., 2018). The number of species predicted to be inhibited is shown. Test-set compounds with experimentally determined, or predicted, broad-spectrum activity ($N. inhibited\ strains \geq 10$) are shown.

Below, we also explore the overlap between the compounds predicted to have broad spectrum activity with the ground truth (panel a). Here, we observe once again that the model based on Chemical Descriptors predicts many false positives, while the ECFP4 model fails to recover many examples. Finally, we explore areas of the chemical space where these models run into trouble (panel b). The area described by human-targeted drugs in the upper right corner has many false positive predictions for the Chemical Descriptor model. The ECFP4 model rarely predicts broad-spectrum activity for compounds not targeted at bacteria or fungi, even when compounds are found in a similar space to known examples (lower right). The figure below is included as Supplementary Figure 5.

We have added these observations in the main text of our manuscript (page 8, l. 91-100):

“To further highlight the superior performance of MoIE-based predictions, we focused on the 24 compounds in the test set that had experimentally determined broad-spectrum antimicrobial activity (i.e., inhibiting the growth of ten or more strains [24]). Figure 3g reports the true number of species affected by each compound (last row) vs. Chemical Descriptor-based, ECFP4-based, and MoIE-based predictions, respectively (top three rows). On average, MoIE-XGBoost achieved the highest accuracy in recovering the measured antimicrobial activity of compounds independent of the intended target of the compound. The ECFP4-based model failed to recall the activity of most human-targeting drugs, while the Chemical Descriptor-based model generally overestimated activities, leading to many false positive predictions (Figure 3g and Supplementary Figure 5a). MoIE-XGBoost recovered five compounds with broad-spectrum activity, not recognized by the ECFP4-based model.”

a

b

Comparing model test-set predictions. **a.** Venn diagram of compounds predicted to have broad-spectrum activity with the ground truth. **b.** UMAP embedding of MoIE's representation of test-set molecules. Predictions of each model are shown, indicating True Positives (TP), False Positives (FP), True Negatives (TN) and False Negatives (FN). The original target species of each compound is indicated by shapes.

5. Figure 4. How does the chemical space of the 2,327-compound library compare to the chemical space of the 1,197-compound library?

We thank the reviewer for this question. Below, we show a UMAP embedding of MoIE's representation of the Maier, et.al. (used for model training) and the MedChemExpress (MCE) subset (the orthogonal library). Overall, the MCE library covers a broader chemical space than the Maier library. Even so, our model was able to recover examples of chemicals that are reported in the literature as having growth-inhibiting activity despite being structurally distinct to the training set. This is more closely examined for the compounds that were selected for experimental validation, as can be seen in response to Reviewer #1's Question #2. Therefore, the MoIE-XGBoost model is shown to be able to generalize to a novel chemical space. The figure below is included as Supplementary Figure 6.

UMAP embedding of MolE's representation the Maier et.al. and MedChemExpress (MCE) libraries. Compounds are colored, indicating the library in which they are present. Non-antibiotic MCE compounds correctly predicted by MolE-xgboost to have antimicrobial activity are highlighted in blue.

6. There should be more elaboration in the main text on how "antimicrobial potential" is calculated and used to guide downstream molecule prioritization.

We thank the reviewer for this suggestion. We restructured the main text accordingly and highlighted how the AP scores were used as one key criterion to select compounds for experimental testing (see page 8, l.207-218). Other criteria are commercial availability and structural diversity (which we also quantified following Reviewer #1's suggestion).

7. The MolE-XGBoost model is applied to the 2,327-compound library (Figure 4a), and five antibacterial predictions are tested in the laboratory against three Gram-negative pathogens and one Gram-positive pathogen. However, this model is trained using 40 human commensal strains. The authors need to discuss in greater detail the phylogenetic relationship between these four pathogens and the commensal isolates from the training dataset.

We thank the reviewer for this interesting suggestion. We have constructed a taxonomic tree of the species used in our study, indicating which are used for model training and which are used for experimental validation. It can be seen that our selected Gram-negative pathogens are in the same phylum, Proteobacteria, which is represented by the *E. coli* strains ED1a and IA11 (the latter also used in experiments), *Bilophila wadsworthia* in our training data. At the same time, *S. aureus* is a member of the Firmicutes phylum, which is represented by a larger number of species in our training set.

Taxonomic tree of the bacterial strains used in this study. Gram stain is shown by tip shape. Strains present in the Maier et al. dataset and used for model training are shown in black. Strains used for experimental validation are shown in pink.

It is worth mentioning that our choice of bacterial strains aimed to cover a wide range of Gram-negative and Gram-positive pathogens of current concern - most of them part of the ESKAPE list (<https://www.dzif.de/en/glossary/eskape>). The selection, therefore, was not based on taxonomy, but rather on the potential of application. We did decide to include a commensal *E. coli* strain as a representative member of the human gut microbiome. We have included more details for our rationale of strain selection in the main text. Nevertheless, a future endeavor (as highlighted in the Discussion) is to include information about the individual bacteria (including taxonomic or phylogenetic relatedness) in future prediction frameworks.

The figure above is included as Supplementary Figure 10. We have updated the main text in our manuscript to reflect these results (page 10, l.259-265):

“We screened these compounds against a panel of bacterial strains that were selected to cover Gram-negative and Gram-positive pathogens, most part of the ESKAPE group [47]. These included the Gram-negatives *Escherichia coli* UTI, *Klebsiella pneumoniae*, *Pseudomonas aeruginosa*, and the Gram-positive *Staphylococcus aureus*, all of which were not present in the training dataset [24]. We also included one shared commensal strain, namely *E. coli* IA11 as a representative of the human gut microbiome. The Gram-negative pathogens were taxonomically related to microbes from the Proteobacteria phylum in the training data, whereas *S. aureus* belonged to the Firmicutes phylum (Supplementary Figure 10).”

8. The antibacterial efficacies of opicapone and ebastine are minimal and, in my opinion, should not be considered antibacterial.

The reviewer correctly points out that Opicapone and Ebastine do not have a *bactericidal* effect on *S. aureus*. That being said, they do affect microbial growth. This sort of finding can be found in our training data as well, where compounds severely limit growth compared to standard growth conditions while not always being bactericidal.

9. Figure 4 and Figure 5 – it would be a tremendously useful addition to the paper to perform the same set of experiments (predictions and in vitro validations) using a model trained using ECFP4 representations as a common method to vectorize molecules. This would serve as the ideal comparison of MolE representations in the context of wet lab validation of model predictions using different molecular representations.

We thank the reviewer for suggesting this interesting analysis. Below, we have re-created Figure 4 using the predictions of the ECFP4-XGBoost model. We observe that this model can also recover the antimicrobial activity of various molecules, most of them being antibiotics (panel a). However, the total number of compounds predicted to have broad-spectrum activity is lower than that of the MolE-XGBoost model (82 vs 158, panel b), and a lower proportion of them are found to have any kind of growth-inhibiting activity against microbial life in the literature (panel b). That being said, this model does reflect more activity against Gram-positive bacteria. The figure below is included as Supplementary Figure 7.

Predicting antimicrobial activity using ECFP4. **a.** UMAP embedding of ECFP4's representation of the 2,320 compounds for which predictions are made. Compounds predicted to inhibit 10 or more strains are highlighted in pink. **b.** Literature-reported activity of 82 non-antibiotic compounds predicted to inhibit at least 10 strains. **c.** Ranking of compounds based on Antimicrobial Potential scores. The predicted number of inhibited strains is compared to Antimicrobial Potential. Known antibiotics are shown in red. **d.** Comparison of the Antimicrobial Potential for Gram-positive and Gram-negative strains determined for non-antibiotic drugs with predicted broad-spectrum activity. Names shown in bold were selected for experimental validation. Color legend is same as in panel b.

A more direct comparison of our MoIE- and ECFP4-based models can be seen in the figure below. In general, both models are able to rank known antibiotics highly in terms of the number of strains predicted to be inhibited, but the ECFP4 model does rank more antibiotics poorly (left tail in panel a). Similarly, the MoIE model is able to recognize, and rank highly, a greater number of non-antibiotic compounds with confirmed antimicrobial activity (panel b). A greater proportion of the non-antibiotic compounds with predicted broad-spectrum activity are confirmed in the literature to inhibit the growth of bacteria, and other forms of microbial life (panel c). Finally, we can see that, when considering non-antibiotic compounds with known antimicrobial activity, a greater number of compounds are found by the MoIE-XGBoost model (panel d). This is included as Supplementary Figure 8.

Comparing predictions of antimicrobial activity on the MCE library. **a.** Histogram of the predicted number of inhibited strains for known antibiotics by ECFP4 (blue) and MoIE (pink) models. **b.** Predicted number of inhibited strains for non-antibiotic compounds with predicted, and confirmed, antimicrobial activity. **c.** Comparison of literature search results for non-antibiotic compounds with predicted broad-spectrum activity. **d.** Comparison of the predicted number of inhibited strains for non-antibiotic compounds with confirmed antimicrobial activity. 13 compounds are uniquely recovered by the ECFP4 model, while MoIE is able to uniquely recover 58. Both methods correctly recover the antimicrobial activity of 24 non-antibiotic compounds.

We have updated the main text to include these observations (page 9-10, I.240-247):

“For completeness, we performed the same analysis using AP scores derived from the ECFP4-XGBoost model. Briefly, AP scores from the ECFP4-XGBoost model deemed (i) fewer compounds to have broad-spectrum activity (Supplementary Figure 7), (ii) fewer known antibiotics to be inhibitory (Supplementary Figure 8a), and (iii) only 82 non-antibiotic compounds to have broad-spectrum activity (vs. 158 from MolE-XGBoost AP scores). Of these, a lower proportion were found to have growth-inhibiting activity against microbial life in the literature (Supplementary Figure 8c). MolE-XGBoost AP scores recovered more non-antibiotic compounds with confirmed activity, 58 of which were not prioritized by ECFP4-XGBoost AP scores. ECFP4-XGBoost prioritized 13 compounds that are not present in the MolE-XGBoost-derived set (Supplementary Figure 8d)”

10. Discussion – there is a reference to “Supplementary Table ??”.

Thank you! We fixed this typo.

Response letter to reviewer comments for Manuscript **NCOMMS-24-40179-A**

“Pre-trained molecular representations enable antimicrobial discovery”

Once again, we would like to sincerely thank the editor and the reviewers for their time and effort in providing valuable feedback regarding our manuscript. The suggestions made led to significant improvements in our work. We are glad that we were able to address all of the concerns of Reviewer #1, and thank them for their comments.

To address the remaining concerns of Reviewer #3, we have improved the clarity of the communication of our original results and conducted an additional analysis, to show that our mode *does* identify powerful antimicrobial compounds in the orthogonal chemical library, and can appropriately rank compounds based on their MIC. As a result, we have updated the manuscript and created new Figures to reflect these new findings.

REVIEWER COMMENTS

Reviewer #1 (Remarks to the Author):

Thank you to the authors for their detailed responses and the additional experiments. I have also reviewed the questions and replies from other reviewers, and I believe my main concerns have been well addressed. Good luck.

We thank the reviewer for their time and effort in reviewing our work. Their valuable comments led to an improvement of the manuscript. Thanks!

Reviewer #3 (Remarks to the Author):

“The manuscript is improved and I thank the authors for their efforts. There are potential benefits to MolE representations over conventional ECFP4 fingerprints, although much room for improvement exists.”

We thank the reviewer for their comments and appreciation for our work. Indeed, we believe that it is an interesting area of research that merits further studies. We address the remaining concerns below.

“My remaining major concern is in the experimental validation of model predictions (Figure 5). Neither opicapone or ebastine would be classified as a reasonable antibiotic given the lack of an observable MIC at any concentration shown in the manuscript. I understand that growth is suppressed, but at no concentration is complete growth inhibition observed. This does, unfortunately, limit the impact of the study.

The authors need to provide detailed explanations as to why more potent antibiotics (at least molecules that have an MIC) weren't identified. Moreover, an obvious question that could be addressed is, would another ML method (perhaps not using MolE representations) predict more potent molecules (defined by in vitro MIC) from this same chemical set? This ultimately defines the true utility of a molecular property predictor.”

We thank the reviewer for their comments. We have approached the remaining concerns in two ways in the manuscript, and provide additional direct comments regarding the experimental results (now Figure 5) below as well.

- 1) The first change in the manuscript is a re-design of Figure 4 that hopefully facilitates a better summary of the types of compounds we find, including many potent molecules (that the model has NOT seen but are known from literature).
- 2) Secondly, we conducted an extensive literature search and collected all available **experimentally measured MICs** for highly ranked compounds from our discovery library, both for MolE- and ECFP-based models. We show that only our MolE-based model shows a meaningful association between experimentally measured MICs and our predictive AP scores (see below for details).

Reply to the concerns regarding our experimental results

We do agree that Opicapone and Ebastine could not be directly used as antibiotics in their current form. Nevertheless, we consider their growth-limiting effects as valuable discoveries. These compounds can serve as a starting point from which chemical analogs can be evaluated for increased potency, a strategy that has already led and continues to lead to the discovery of novel antibiotics (<https://www.nature.com/articles/s41570-021-00313-1>, Ref. [1] in the manuscript). Independently, these

experimental findings continue to shed light on the growth-inhibiting effects of non-antibiotic drugs and can be used as training data in future studies.

We also would like to re-iterate that, for experimental validation, we **deliberately focused on evaluating non-antibiotic compounds** that had **no previously reported antibacterial activity**. This contributes to added difficulty in finding novel growth-inhibiting compounds. Even so, we were able to identify the powerful activity of **Elvitegravir**. We still believe that (i) these results support the utility of MolE's molecular representation for appropriately identifying and ranking powerful antimicrobial compounds and (ii) our pre-train/fine-tune/experimental validation workflow is a step forward in the automatic antimicrobial discovery process.

Reply to the concerns regarding finding potent molecules

To facilitate a better summary of the potency of certain molecules and our findings in general, we updated Figure 4 (see further below) and provide additional remarks here. First of all, we would like to highlight that our model-derived Antimicrobial Potential scores **do identify many potent molecules** with an observable MIC and, indeed, rank them highly in terms of AP scores. Briefly, we performed **de novo** predictions on an orthogonal (or "discovery") chemical library from MedChemExpress (MCE) **without prior knowledge** of the outcomes. This means, MolE-XGBoost has **never** seen these molecules. Moreover, the molecules in the discovery library were chemically more diverse than the Maier et al. training set (measured in terms of joint UMAP embeddings, see Supplementary Figure 6, included below).

Supplementary Figure 6: A UMAP embedding of the MolE representation of the compounds in the MedChemExpress (MCE) and Maier et. al. libraries. Some examples of literature-validated predictions of antimicrobial compounds from MCE library are highlighted.

We then examined the compounds our model predicted to inhibit $K \geq 10$ or more strains. In our updated Figure 4 (seen below), it can now be better appreciated that **33% of the compounds** highlighted by MoIE-XGBoost are, in fact, known antibiotics (panels b and c), and that these compounds are also ranked highly in terms of their Antimicrobial Potential (AP) score G . Furthermore, among the **non-antibiotic compounds** highlighted by the model, 33% had been previously reported as having growth-inhibiting activity against bacteria (green ring in Figure 4b). Together, these results show that our model identifies highly potent antimicrobial molecules in this discovery library **without prior knowledge**.

Predicting antimicrobial potential in the discovery MCE-based chemical library comprising 2,320 compounds. **a.** UMAP embedding of MoIE's representation of the 2,320 compounds for which predictions are made. Compounds predicted to inhibit at least 10 strains ($K \geq 10$) are highlighted in blue. **b.** Literature-reported categorization of the antimicrobial activity for the 235 compounds with $K \geq 10$. This set comprises 77 antibiotics (33%, shown in red) and 158 non-antibiotic drugs (67%, shown in blue). The non-antibiotic drugs are further categorized into five classes (colored in the outer ring). **c.** Antimicrobial Potential score G vs. number of predicted inhibited strains K of all 2,320 compounds. Known antibiotics ($n = 93$) are shown in red. The dashed line marks $K = 10$. Non-antibiotic compounds with $K \geq 10$ ($n = 158$) are shown in blue. **d.** Scatter plot of the Antimicrobial Potential for Gram-positive (G^+) vs. Gram-negative strains (G^-) determined for the 158 non-antibiotic drugs with predicted broad-spectrum activity ($K \geq 10$). The coloring of each compound corresponds to the categorization in panel b. Six compounds with names shown in bold were selected for experimental validation.

Addition of a new Results section: Quantitative analysis between experimentally determined MICs and AP scores

To further validate the ability of our model, we conducted an additional analysis where we evaluated the ability of the Antimicrobial Potential score to correlate with reported *in vitro* MICs. To do this, we performed a deeper literature search and gathered the *in vitro* MICs reported for the **non-antibiotic compounds** predicted by our model to have **broad-spectrum activity** ($K \geq 10$). We found 31 compounds with a $MIC \leq 128 \mu\text{g/mL}$ against Gram-positive and -negative species. In the figure below

(new Figure 5 in the manuscript), it can be observed that the Antimicrobial Potential score **significantly anti-correlates** with the magnitude of the experimentally determined MIC, meaning the compounds with a **higher Antimicrobial Potential** have a more potent **growth-inhibiting activity** (panel a). We believe that this is a strong confirmation that our model can meaningfully rank antimicrobial compounds with low *in vitro* MICs. This analysis also supports the generalizability of our model to a wide range of bacterial species, as most of the MICs gathered were determined for **species not present in our training dataset**.

Figure 5 (new): Relationship between AP scores and minimum inhibitory concentration (MIC). **a.** Regression and correlation analysis between the AP score G and the corresponding log₂ literature-reported MICs ($\mu\text{g}/\text{mL}$) for 31 non-antibiotic compounds against Gram-positive (shown in blue) and Gram-negative (shown in yellow) species (39 compound-species combinations in total, Spearman's $\rho = -0.5$). The linear regression fit \pm standard error is shown in pink. The slope of the regression is annotated. **b.** AP scores (AP-G) and MIC ($\mu\text{g}/\text{mL}$) values of the compounds with top-3 and bottom-3 MIC values along with the respective bacterial species inhibited

Importantly, when performing the same analysis for the model trained with **ECFP4**, we found **no significant relationship** between the **ECFP4-based** Antimicrobial Potential and the reported MIC (figure below, panel a). Overall, we found a greater number of compounds with a MIC $\leq 128 \mu\text{g}/\text{mL}$ when using the model trained with MoLE (panel b). This includes more non-antibiotic compounds active against Gram-negative and Gram-positive species (panel c)

Part of Supplementary Figure 9 (new): Antimicrobial Potential scores and reported MIC. **a.** Relationship between the AP-score G of the ECFP4 model and the \log_2 MIC for the 12 non-antibiotic compounds ($K \geq 10$, according to ECFP4-XGBoost) with a reported MIC $128 \mu\text{g}/\text{mL}$ against any bacterial species (17 compound-species combinations in total). A linear regression fit \pm standard error is shown in blue. The slope of the regression along with its p-values is shown. **b.** The number of compounds with a reported MIC $\leq 128 \mu\text{g}/\text{mL}$, recovered by MoIE and ECFP4. Non-antibiotic and antibiotic compounds are shown. **c.** The number of compounds with a MIC $\leq 128 \mu\text{g}/\text{mL}$ against Gram-positive and Gram-negative species.

Reply to the comment regarding another ML method

Certainly, there could be (and likely will be) other ML methods that could potentially outperform XGBoost. We have tested many methods, including standard Random Forest and fine-tuned MLP-NNs. For us, further improving the downstream pipeline and comparing with additional representations is an active area of future research, and we hope that the extensive comparisons we provide here, the reproducible software and data provided within this manuscript, can indeed kickstart such an endeavor (in the same vein as how MoleculeNet did facilitate molecular learning).

Reviewer #3 (Remarks on code availability):

No issues encountered.

We thank the reviewer for their efforts in evaluating our codebase. The suggestions improved usability considerably.